# Prevalence of persistent SARS-CoV-2 in a large community surveillance study

Mahan Ghafari[1,2,3 ✉], Matthew Hall[1,3], Tanya Golubchik[1,4], Daniel Ayoubkhani[5,6], Thomas House[7], George MacIntyre-Cockett[1,8], Helen R. Fryer[1], Laura Thomson[1,3], Anel Nurtay[1], Steven A. Kemp[1,2,3], Luca Ferretti[1,3], David Buck[8], Angie Green[8], Amy Trebes[8], Paolo Piazza[8], Lorne J. Lonie[8], Ruth Studley[5], Emma Rourke[5], Darren L. Smith[9,10], Matthew Bashton[9,10], Andrew Nelson[10], Matthew Crown[9,10], Clare McCann[10], Gregory R. Young[9,10], Rui Andre Nunes dos Santos[10], Zack Richards[10], Mohammad Adnan Tariq[10], Roberto Cahuantzi[5], Wellcome Sanger Institute COVID-19 Surveillance Team[11], COVID-19 Infection Survey Group*, The COVID-19 Genomics UK (COG-UK) Consortium*, Jeff Barrett[12], Christophe Fraser[1,3,8,12], David Bonsall[1,3,8,13], Ann Sarah Walker[14,15,16,17] & Katrina Lythgoe[1,2,3 ✉]

Persistent SARS-CoV-2 infections may act as viral reservoirs that could seed future outbreaks[1–5], give rise to highly divergent lineages[6–8] and contribute to cases with post-acute COVID-19 sequelae (long COVID)[9,10]. However, the population prevalence of persistent infections, their viral load kinetics and evolutionary dynamics over the course of infections remain largely unknown. Here, using viral sequence data collected as part of a national infection survey, we identified 381 individuals with SARS-CoV-2 RNA at high titre persisting for at least 30 days, of which 54 had viral RNA persisting at least 60 days. We refer to these as 'persistent infections' as available evidence suggests that they represent ongoing viral replication, although the persistence of non-replicating RNA cannot be ruled out in all. Individuals with persistent infection had more than 50% higher odds of self-reporting long COVID than individuals with non-persistent infection. We estimate that 0.1–0.5% of infections may become persistent with typically rebounding high viral loads and last for at least 60 days. In some individuals, we identified many viral amino acid substitutions, indicating periods of strong positive selection, whereas others had no consensus change in the sequences for prolonged periods, consistent with weak selection. Substitutions included mutations that are lineage defining for SARS-CoV-2 variants, at target sites for monoclonal antibodies and/or are commonly found in immunocompromised people[11–14]. This work has profound implications for understanding and characterizing SARS-CoV-2 infection, epidemiology and evolution.

The emergence of highly divergent variants of SARS-CoV-2 has been a defining feature of the COVID-19 pandemic. Although the evolutionary origins of these variants are still a matter of speculation, multiple pieces of evidence point to chronic persistent infections as their most likely source[5,7,15]. In particular, infections in immunocompromised patients who cannot clear the virus may lead to persistence for months[6,7,16,17] or even years[8,18] before potentially seeding new outbreaks in the community[3]. Persistence of SARS-CoV-2 during chronic infections exposes the viral population to host immune responses and other selective pressures as a result of treatments over prolonged periods of time. Persistent infections also release the virus from undergoing the tight population bottlenecks that are characteristic of SARS-CoV-2 transmission[19,20], making the viral population less vulnerable to stochastic genetic drift and allowing it to acquire more evolutionary changes over a longer timescale. These adaptive intra-host changes can lead to elevated evolutionary rates, particularly in key regions of the spike

[1]Big Data Institute, Nuffield Department of Medicine, University of Oxford, Oxford, UK. [2]Department of Biology, University of Oxford, Oxford, UK. [3]Pandemic Science Institute, University of Oxford, Oxford, UK. [4]Sydney Infectious Diseases Institute (Sydney ID), School of Medical Sciences, Faculty of Medicine and Health, University of Sydney, Sydney, New South Wales, Australia. [5]Office for National Statistics, Newport, UK. [6]Leicester Real World Evidence Unit, Diabetes Research Centre, University of Leicester, Leicester, UK. [7]Department of Mathematics, University of Manchester, Manchester, UK. [8]Wellcome Centre for Human Genetics, Nuffield Department of Medicine, NIHR Biomedical Research Centre, University of Oxford, Oxford, UK. [9]The Hub for Biotechnology in the Built Environment, Department of Applied Sciences, Faculty of Health and Life Sciences, Northumbria University, Newcastle upon Tyne, UK. [10]Department of Applied Sciences, Faculty of Health and Life Sciences, Northumbria University, Newcastle upon Tyne, UK. [11]Wellcome Sanger Institute COVID-19 Surveillance Team, https://www.sanger.ac.uk/project/wellcome-sanger-institute-covid-19-surveillance-team/. [12]Wellcome Sanger Institute, Cambridge, UK. [13]Oxford University Hospitals NHS Foundation Trust, John Radcliffe Hospital, Headington, Oxford, UK. [14]Nuffield Department of Medicine, University of Oxford, Oxford, UK. [15]The National Institute for Health Research Health Protection Research Unit in Healthcare Associated Infections and Antimicrobial Resistance at the University of Oxford, Oxford, UK. [16]The National Institute for Health Research Oxford Biomedical Research Centre, Oxford, UK. [17]MRC Clinical Trials Unit at UCL, UCL, London, UK. *Full lists of names and affiliations for the COVID-19 Infection Survey Group and the COG-UK Consortium members are provided in the Supplementary Information. ✉e-mail: mahan.ghafari@ndm.ox.ac.uk; katrina.lythgoe@biology.ox.ac.uk

protein (encoded by *S*) that are often associated with immune escape and increased rates of transmission[13,14].

Despite the substantial public health implications of persistent infections, uncertainty still surrounds how common these infections are among the general population, how long they last, their potential for adaptive evolution and their contribution to long COVID.

In this work, we used genetic, symptom and epidemiological data from the Office for National Statistics COVID Infection Survey (ONS-CIS)[21], a large-scale community-based surveillance study carried out in the UK. We identified individuals with high-titre SARS-CoV-2 samples spanning 1 month or more and representing the same viral population. We have provided several lines of evidence suggesting that these individuals are persistently infected with replicating virus, and hence refer to these as persistent infections; however, the presence of non-replicating SARS-CoV-2 RNA cannot be categorically ruled out in all cases. We characterized various aspects of viral dynamics during these persistent infections, including evolutionary changes in the virus, RNA viral titre kinetics (hereafter referred to as viral load), number of reported symptoms and prevalence of long COVID, the last in comparison with individuals without identified persistent SARS-CoV-2 infection.

## Identifying persistent infections

We considered 93,927 high-quality sequenced samples from the ONS-CIS collected between 2 November 2020 and 15 August 2022, and representing 90,146 people living in 66,602 households across the UK (see Extended Data Fig. 1). Households representative of the UK population were recruited in the survey using a rolling recruitment strategy. Most participating individuals (approximately 98%) were sampled once a week for the first 4 weeks of their enrolment, and then approximately monthly thereafter, regardless of symptoms or testing history. To identify persistent infections, we first limited the dataset to individuals with two or more PCR with reverse transcription (RT–PCR)-positive samples with cycle threshold (Ct) values ≤ 30 in which sequencing was attempted (a proxy for viral load), taken at least 26 days apart, and where the consensus sequences were of the same major lineages of B.1.1.7 (hereafter referred to as Alpha), B.1.617.2 (hereafter referred to as Delta), or the two Omicron lineages BA.1 or BA.2 (BA.4, BA.5 and XBB were not considered). This included a total of 500 individuals (18 Alpha, 122 Delta, 130 BA.1 and 230 BA.2) with two or more sequences of the same major lineage (including those with at least one undetermined lineage; see Extended Data Table 1). If sequences from the same individual also shared the same rare single-nucleotide polymorphisms (SNPs) at one or more sites relative to the major-lineage population-level consensus, we classified them as having a persistent infection. Because we used sequence data to identify persistent infections, we could only identify persistent infections with at least two high viral load (Ct ≤ 30) samples.

We defined a rare mutation for a given lineage as one observed in 400 or fewer samples of that lineage within the ONS-CIS dataset, giving a false-positive rate of identifying persistent infections of 0–3% depending on the major lineage (see Methods; Extended Data Fig. 2). We note that the rare SNP method provides a conservative estimate for the true number of persistent infections, as some persistent infections may not have rare mutations. To evaluate the robustness of our method for identifying persistent infections, we considered the phylogenetic relationship between the sequences from persistent infections relative to other sequences of the same major lineage that belonged to individuals with only a single sequence within the ONS-CIS dataset. The great majority of sets of sequences identified as belonging to the same persistent infection formed monophyletic groups with strong bootstrap support (Fig. 1a and Extended Data Fig. 3). However, seven sequences did not group with the other sequence (or sequences) from the same persistent infection. All of these had high Ct values (Ct ≈ 30) and low genome coverage, which could explain their lack of clustering on the

phylogeny as lower-coverage sequences are more likely to lack information at lineage-defining sites and they may be more prone to errors when calling the consensus[19]. In particular, two of these sequences were collected at intermediate time points of two persistent infections where the first and last sequences of each persistent infection do cluster on the phylogeny, whereas the sequence at the intermediate time point does not.

We identified 381 persistent infections with sequences spanning at least 26 days (11 Alpha, 106 Delta, 97 BA.1 and 167 BA.2). The relatively low number of persistent infections that we identified for Alpha is probably because fewer individuals were infected with Alpha than the other major lineages, but also because a smaller proportion of positive samples with Ct ≤ 30 were sequenced before December 2020, which captures the beginning of the Alpha wave, than after this date (see supplementary figure S1 in ref. 22). Of all the persistent infections that we identified, 54 spanned at least 56 days (3 Alpha, 13 Delta, 15 BA.1 and 23 BA.2). This represents nearly 0.07% (54 of 77,561) of all individuals with one or more sequences (with Ct ≤ 30) of the four major lineages that we investigated in this study (Fig. 1b; see also Table 1). Of note, 2 Alpha, 19 Delta and 8 BA.1 persistent infections were sampled weeks after the corresponding major lineage had dropped to a frequency of 1% or less (Fig. 1c,d); the longest infection was with BA.1 and lasted for at least 193 days (see Fig. 1b).

The actual duration of persistent infections is likely to be at least 3–4 days longer than the time between when the first and last sequenced samples were collected, as it typically takes 3–4 days since the start of infection for viral loads to be sufficiently high to be sequenced (Ct ≤ 30)[23,24] and, similarly, viral loads will be too low (Ct values too high) to sequence at the tail end of infection. As individuals were typically sampled weekly during the first 4 weeks of enrolment, followed by approximately monthly sampling thereafter, it is unsurprising that most persistent infections had observable durations clustering around 30 or 60 days (Extended Data Fig. 4).

## Identifying reinfections

We considered a pair of sequences from the same individual to indicate a reinfection with the same major lineage if they were sampled at least 26 days apart, had at least one consensus nucleotide difference between the sequenced sampling time points and shared no rare SNPs (see Methods). This criterion may overestimate the true number of reinfections with the same major lineage as some persistent infections may not have a rare SNP, and within-host evolution can lead to the loss of a rare SNP and/or the gain of other mutations leading to differences in the consensus sequence between the samples. We cannot rule out samples being attributed to the wrong individuals, which would also overestimate the true number of reinfections, although we took several measures to control for sample mix-ups (see Methods). We identified three individuals for which pairs of sequences from different sampling time points had no identical rare SNPs and at least one consensus difference, but whose viral load trajectories were consistent with a persistent chronic infection. We therefore excluded these individuals from the reinfection group (Extended Data Fig. 5).

Overall, we identified 60 reinfections with the same major lineage (7 Alpha, 11 Delta, 14 BA.1 and 28 BA.2; Table 1). Sequences from individuals identified as reinfected, collected at the point of primary infection and reinfection, did not form monophyletic groups and mostly belonged to distantly related subclades, and hence supports our method for identifying reinfections (Fig. 1a and Extended Data Fig. 3).

Of all the cases classed as either persistent infections or reinfections with the same major lineage, 9–39% were classed as reinfections (Table 1), rising to 12–50% if only samples collected at least 56 days apart were included (Fig. 1b). This suggests that for Delta, BA.1 and BA.2, the number of individuals reinfected with the same major lineage is low compared with the number of individuals with persistent infection.

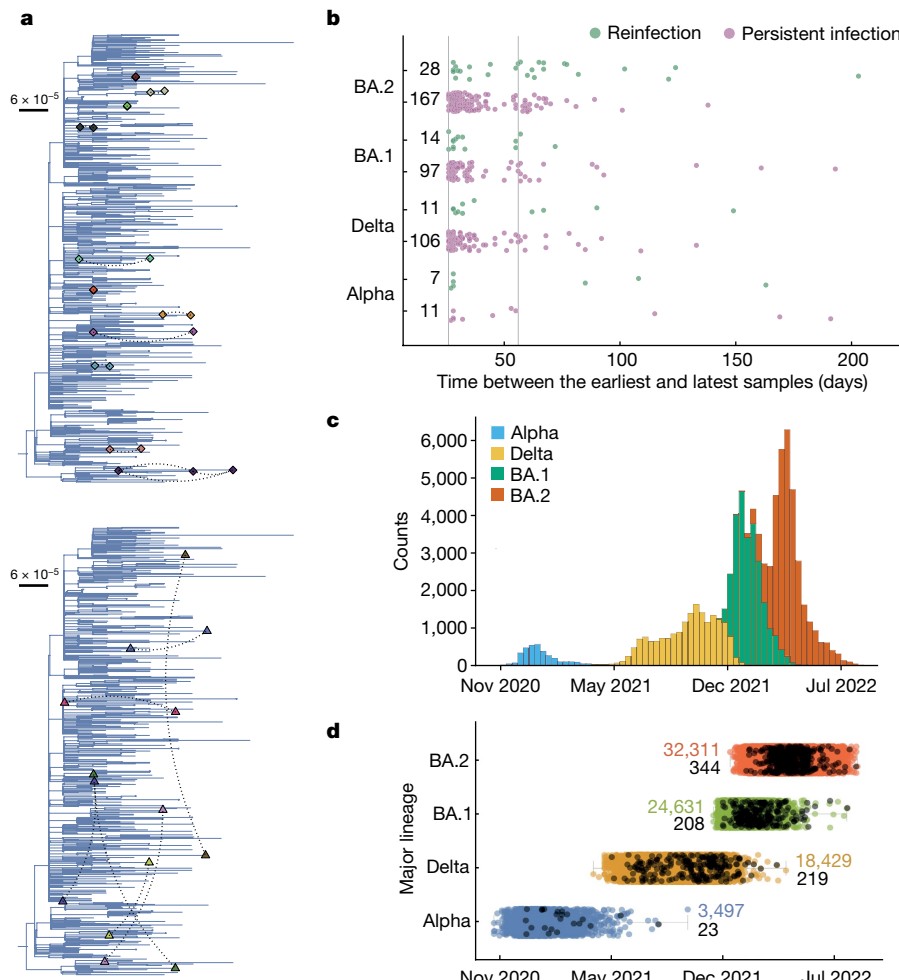

**Fig. 1 | Individuals identified with persistent SARS-CoV-2 and reinfections with the same major lineage within the ONS-CIS. a**, Phylogenetic relationship between samples from individuals with persistent SARS-CoV-2 RNA (hereafter referred to as persistent infections) (top), and reinfections with a representative background population of Alpha (B.1.1.7; see Extended Data Fig. 2 for the analysis on the other three major lineages) (bottom). The dashed lines connect every pair of sequences from the same individual. Pairs from individuals with persistent infections cluster closely together, whereas reinfections do not. All sequences from the same individual are given the same colour. **b**, Number of days between the earliest and latest genomic samples from persistent infections and reinfections. Each point represents a single individual. The solid vertical lines show the 26-day and 56-day cut-offs. The numbers on the side of each bar show the total counts per category for each major lineage. **c**, Total

number of sequences in the ONS-CIS per major lineage over time. **d**, Timing of persistent infections (black) during the UK epidemic. Some individuals with persistent infections can be identified up to weeks after the lineage has been replaced at the population level. The coloured boxes indicate the interquartile range, which spans from the 25th to the 75th percentile, with the centre being the median calendar date corresponding to each major lineage. The medians for Alpha, Delta, BA.1 and BA.2 are 13 January 2021, 16 October 2021, 20 January 2022 and 30 March 2022, respectively. The extremities (displayed as grey horizontal lines) denote the minimum and maximum values within each category. The coloured numbers on the side of each box show the total number of sequences within the ONS-CIS for each major lineage. The black numbers represent the total number of sequences from persistent infections corresponding to each major lineage.

Alpha seems to be an exception with over one-third of cases classed as reinfection for samples that were 26 days or more and half of cases for samples that were 56 days or more apart. This may be because of the lower number of Alpha samples sequenced, but other factors such as the timing of vaccination roll out could also have contributed.

## Periods of stasis at the consensus level

Of the 381 persistently infected individuals that we identified, 68% (259 of 381) displayed no nucleotide differences at the consensus level during infection. By contrast, when we determined the number of consensus nucleotide differences between 16,000 random pairs of sequences from the ONS-CIS, and with each pair from the same major lineage, only 6 pairs had no consensus differences (Extended Data Fig. 6). This provides further support that the sequences that we identified from persistent infections belong to the same infection.

The lack of consensus changes between many pairs of samples taken from the same infection, most of which are less than 2 months apart, is consistent with neutral evolution or weak selection, and indicates that there was limited within-host adaptation. In support of this, we identified 17 persistent infections with three or more sequences, of which the first two sequences (typically about 1 month apart) had zero consensus differences, but, crucially, 41% (7 of 17) gained a consensus change later in the infection. This suggests that the virus evolves measurably at the consensus level as time progresses since the onset of infection. However, shifting populations of RNA-producing cells (and sampling differences) could also potentially lead to differences in the consensus between different time points in the absence of ongoing replication. Among the remaining 59% (10 of 17) with no consensus change throughout infection, we often found substantial sub-consensus activity with intra-host single-nucleotide variant frequencies going up to high frequencies (approximately 40%) and returning to below 5% at a later

**Table 1 | Number of persistent infections and reinfections per major lineage**

| Major lineage | Reinfection 26 days or longer | Persistent infection 26 days or longer | Reinfection (%)[a] 26 days or longer | Reinfection 56 days or longer | Persistent infection 56 days or longer | Reinfection (%)[a] 56 days or longer |
|---|---|---|---|---|---|---|
| Alpha | 7 | 11 | 39 | 3 | 3 | 50 |
| Delta | 11 | 106 | 9 | 4 | 13 | 24 |
| BA.1 | 14 | 97 | 13 | 2 | 15 | 12 |
| BA.2 | 28 | 167 | 14 | 15 | 23 | 40 |

[a]Reinfection (%)=reinfection/(reinfection+persistent infection)×100.

time point, indicating that the virus population is probably replicating during infection despite acquiring no consensus change (Extended Data Fig. 7a,b).

## A strong signal for positive selection

Despite long periods with little or weak positive selection, we also found evidence for positive selection. Among the 381 persistently infected individuals, we observed 317 changes in the consensus nucleotide representing 277 unique mutations and 31 deletions representing 18 unique deletions. Many of these mutations have previously been identified as either lineage-defining mutations for variants of concern or variants of interest[25] (8 mutations and 2 deletions), recurrent mutations in immunocompromised individuals[12–14] (15 mutations and 4 deletions) or key mutations with antibody escape properties and target sites for various different monoclonal antibodies[11,26] (7 mutations) (see Source Data Fig. 2 and Table 2).

Several of the consensus changes that we observed were at the same genomic positions in multiple individuals. For example, three individuals infected with BA.2 from different households acquired a mutation at codon position 547 in the spike protein (Fig. 2), two of which were the T547K mutation, which is a lineage-defining mutation for BA.1, and one the K547T mutation (Table 2; also see Source Data Fig. 2). Twelve individuals acquired a deletion (open reading frame (ORF) 1ab (*ORF1ab*): Δ81–87) in the NSP1-coding region. A similar deletion has previously been observed during the chronic infection of an immunocompromised individual with cancer[16] and has also been associated with lower type I interferon response in infected cells[27].

Overall, we observed a strong signal for positive selection in *S*, with nearly ninefold more non-synonymous than synonymous mutations (Fig. 2b). With a total of seven non-synonymous mutations, *ORF8* had the highest per base (0.036 per base) number of non-synonymous mutations, followed by *S* with 61 non-synonymous mutations (0.016 per base). The high number of non-synonymous mutations in *ORF8* may be due to premature stop codons scattered along *ORF8*, meaning the downstream non-synonymous mutations are released from negative selection[28].

## Frequently observed mutations

We determined the number of times each of the consensus change mutations that we observed during persistent infections appeared on representative global and English phylogenies, and compared this with the number of times any mutations observed on the phylogenies occurred (see Methods; Extended Data Fig. 8a,b). In general, mutations that emerged during persistent infection appeared more frequently on the global and English phylogenies, and with mutations emerging multiple times during persistent infection appearing more frequently still (Extended Data Fig. 8c).

Mutations leading to consensus change during persistent infections also tended to be more beneficial at the population level, where here fitness is defined by their ability to spread among individuals[29], than other mutations found in the global phylogenies of B.1.1.7, B.1.617.2,

BA.1 and BA.2 (Extended Data Fig. 8d). Moreover, mutations observed to appear in multiple persistent infections tended to have a stronger positive fitness effect than those observed in only a single persistent infection (Extended Data Fig. 8e). This indicates that mutations that are selected during persistent infections also tend to be better at transmitting between individuals. Of note, however, are two mutations that emerged twice during persistent infections and were mildly deleterious based on the global phylogeny. These were T1638I (also known to be recurrent in immunocompromised individuals[13]) and T4311I in *ORF1ab*. This suggests that these mutations may be beneficial at the within-host level, at least in some individuals, but deleterious at the between-host level; however, it is important to recognize that the ability of immune escape mutations to spread among individuals could change through time due to the changing immune landscape of the population.

One BA.1 persistent infection particularly stood out. This infection lasted for at least 133 days during which 33 unique mutations (23 mutations in *ORF1ab*, 6 in *S*, 1 in *ORF3a*, 1 in *M* (encoding the membrane protein) and 2 in *ORF7*) were observed (Extended Data Figs. 3 and 7c); 11 of the *ORF1ab* mutations and all of the mutations in *S*, *ORF3a* and *ORF7* were non-synonymous. Contamination could be ruled out because intra-host single-nucleotide variants were shared across multiple time points (Extended Data Fig. 7c), and co-infection is unlikely as we could not identify a likely co-infecting variant after examining all of the ONS-CIS sequences. Given the mutational signature from

**Table 2 | Recurrent mutations and deletions identified during persistent SARS-CoV-2 infections**

| Gene | Mutation | n | Lineage | Description |
|---|---|---|---|---|
| *S* | T547K | 2 | BA.2 | Lineage-defining for BA.1 |
| *S* | L452R | 2 | BA.2 | Lineage-defining for Delta and BA.4/5[a] |
| *S* | T376A | 2 | BA.1 | Lineage-defining for BA.2/4/5 |
| *S* | T95I | 1 | BA.2 | Lineage-defining for BA.1[a] |
| *S* | G446V/D | 2 | Delta and BA.2 | Target for monoclonal antibodies |
| *S* | D215G | 1 | BA.2 | Lineage-defining for Beta[a] |
| *S* | ΔA243/L244 | 3 | Alpha and BA.2 | Lineage-defining for Beta[a] |
| *S* | ΔY144 | 1 | BA.2 | Lineage-defining for Alpha and BA.1 |
| *ORF1ab* | L5905F | 2 | BA.1 and BA.2 | Commonly found in Mu and Delta |
| *ORF1ab* | T4175I | 2 | BA.1 | Commonly found in BA.2 |
| *ORF1ab* | T1638I | 2 | BA.2 | [a] |
| *ORF1ab* | D4532D | 1 | BA.2 | [a] |
| *ORF1ab* | Δ81–86 | 12 | Delta, BA.1 and BA.2 | [a] |
| *ORF8* | I121L | 1 | BA.1 | [a] |

See Source Data Fig. 2 for more information about all mutations. Δ, deletion. [a]Recurrent in immunocompromised patients.

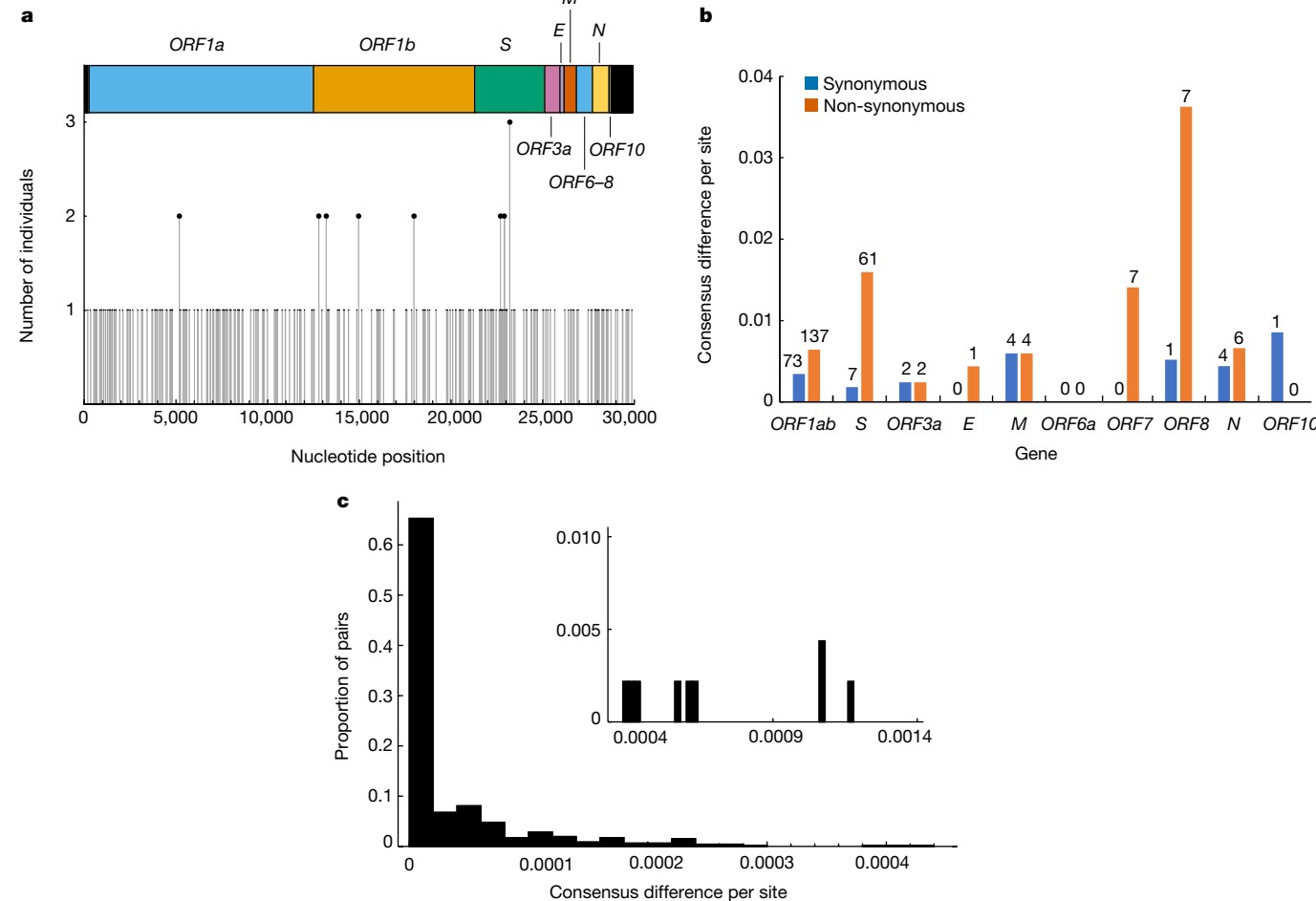

**Fig. 2 | Distribution of SNPs and non-synonymous versus synonymous mutations detected in individuals with persistent SARS-CoV-2. a**, Number of mutations that resulted in a consensus change identified in one or more individuals with persistent SARS-CoV-2 RNA (hereafter referred to as persistent infections). *E*, envelope protein; *M*, membrane protein; *N*, nucleocapsid protein. **b**, Number of synonymous (blue) and non-synonymous (orange) mutations per site during persistent infections. The numbers above each column show the total counts of consensus changes in each category of mutations. **c**, Distribution of consensus differences per site between sequences from all persistent infections. Nearly 65% of all pairs of sequences from the same infection (corresponding to 70% of persistent infections) had zero consensus differences and most others had below 0.0004 differences per site. The inset shows the remaining pairs with a high number of consensus differences.

this persistent infection with 17 G-to-A mutations and very few C-to-A mutations, it is possible that these mutations are induced after a molnupiravir treatment[30].

## Persistence with rebounding viral load

Of the 381 persistent infections, 65 had three or more RT–PCR tests taken over the course of their infection. We classified these infections as 'persistent rebounding' if they had a negative RT–PCR test during the infection ($n = 20$) and the rest as 'persistent chronic' ($n = 47$) (Fig. 3a,b). Given the weekly or approximately monthly sampling of individuals enrolled in the ONS-CIS, infections classed as persistent chronic may have unsampled periods of very low viral load, meaning the persistent-rebounding category is likely to be an underestimate.

Nonetheless, the observation of rebounding viral load dynamics in over 30% of cases is striking given that, in the absence of genetic information, they could have been misidentified as reinfections, depending on the definition used. Of the 27 cases identified as reinfections with three or more RT–PCR tests, all showed rebounding viral load dynamics (Fig. 3c). Also striking is that persistent-chronic infections often showed similar dynamics; of the 47 infections classed as persistent chronic, 35 had a low viral load (high Ct) test between two high viral load (low Ct)

tests. Overall, 55 of 67 (82%) of persistent infections in which we had sufficient data showed a resurgence in viral load after an initial drop (Extended Data Fig. 5a). These rebounding viral load dynamics support the presence of replicating viruses during these infections. There are also several studies that find a strong correlation between high viral load samples (similar to those that we observed here) and the presence of viable SARS-CoV-2 in viral cultures[24,31–33], which further supports that these samples are taken from replication-competent viruses. However, variation in viral load samples may also occur due to reasons unrelated to the presence of replication-competent virus such as variation in measured Ct values with respect to time and quality of sampling[33].

As the sampling strategy of ONS-CIS is based on testing representative individuals across the UK regardless of symptoms, we can estimate the percentage of SARS-CoV-2 infections that are persistent and last for longer than 60 days in the general population. This requires making assumptions about how many persistent infections are missed among ONS-CIS participants due to the approximately monthly (and weekly) sampling. More precisely, estimating the proportion of infections that are persistent depends on the proportion of days the infection has sequenceable virus during the infection (would have Ct ≤ 30 if tested); the fewer the number of days the infection has sequenceable virus, the more likely it is that a persistent infection is missed. By taking two

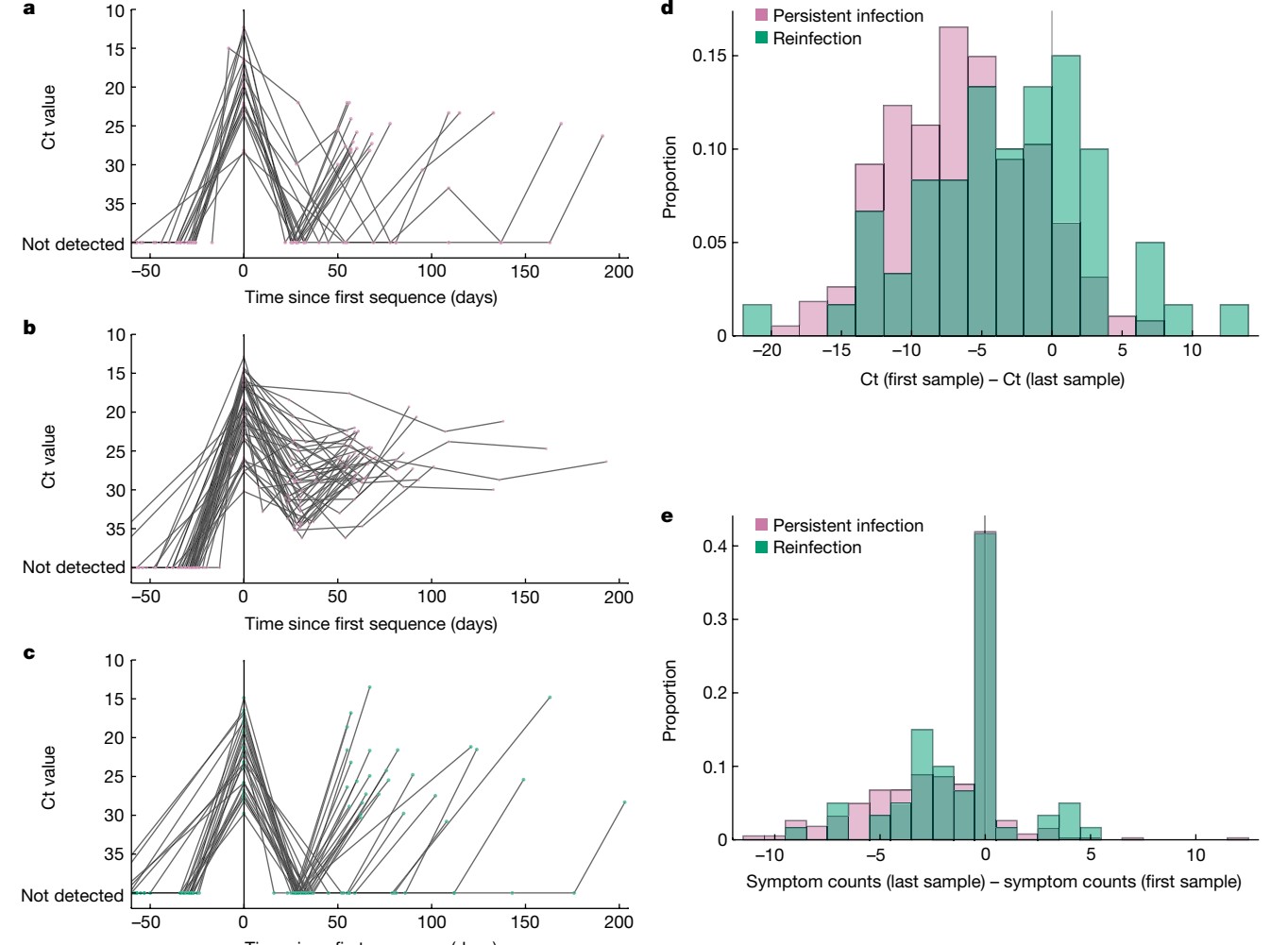

**Fig. 3 | Comparison of RNA viral load dynamics and the number of reported symptoms in individuals with persistent SARS-CoV-2 and reinfections with the same major lineage. a–c**, RNA viral load trajectories of individuals with persistent SARS-CoV-2 RNA (hereafter referred to as persistent infections) with rebounding (that is, a negative RT–PCR test during the infection) (purple; **a**) and chronic persistent viral load (purple; **b**) and reinfections with at least three PCR tests taken over the course of infection or until reinfection (cyan; **c**). For **a–c**, only individuals with three or more RT–PCR tests during the course of infection were included. **d,e**, Change in Ct value (**d**) and total number of symptoms reported between the first and last time points (**e**) with sequenced samples for all 381 persistent infections and 60 reinfections.

extreme scenarios for the proportion of days that the virus is sequenceable during persistent infection (0.7 and 0.14; see Methods), we estimate that approximately 0.7–3.5% and 0.1–0.5% of infections become persistent for more than 30 and 60 days, respectively.

## Difference in viral load and symptoms

For the majority of persistent infections, Ct values (inversely proportional to viral load[34]) were higher at the last sequenced time point than at the first sequenced time point (Fig. 3d), with the Ct value being more than +6.7 (interquartile range (IQR) +3.2–10.2) units higher at the last time point (two-sided paired Student's *t*-test $P = 2 \times 10^{-9}$). For reinfections with the same major lineage, the last sequenced sample also had higher Ct values than the first, but the magnitude of the difference was smaller than persistent infections (Fig. 3d), with only +2.5 (IQR −1.1 to +7.4) units difference between primary infection and reinfections (two-sided paired Student's *t*-test $P = 0.0003$). In both cases, the rise in Ct value (decrease in viral load) during infections or between reinfections could be a consequence of host immunity or within-host compartmentalization. In addition, the rise in Ct for reinfections could be due to the disproportionate sampling of individuals with older

infections, which tend to have lower viral loads, towards the end of an epidemic wave[35,36].

Individuals with persistent infections remained largely asymptomatic during the later stages of infection, reporting on average two fewer symptoms in the preceding 7 days at the last time of sampling (at which a sequence was obtained) than the first time of sampling, with a median of 1 (IQR 0–4) fewer reported symptoms (two-sided paired Wilcoxon $P = 5 \times 10^{-30}$). They also consistently reported very few or no symptoms after the first positive sample (Fig. 3e). In comparison, individuals reinfected with the same major lineage reported on average only one fewer symptom at the reinfection sampling time point than at the primary sampling time point (Fig. 3e), with a median of 0 (IQR 0–3) fewer reported symptoms (two-sided paired Wilcoxon $P = 0.005$). In addition, the proportion of individuals reporting more symptoms at the last sampling is higher among the reinfections than among the persistent infections.

## Prevalence of long COVID

From February 2021, as well as reporting symptoms, participants were asked whether they describe themselves as having long COVID and if

**Table 3 | Prevalence of long COVID in individuals with persistent infection. Individuals with non-persistent infections are set as reference for odds ratio calculations**

| Group | Total *n* | *n* with long COVID (%) | Median follow-up (IQR) | Unadjusted OR (95% CI) | Adjusted OR (95% CI) |
|---|---|---|---|---|---|
| *Long COVID at first assessment 12 weeks or longer post-infection* | | | | | |
| Persistently infected | 356 | 32 (9.0) | 101 (91–113) | 1.27 (1.19–2.47) | 1.55 (1.07–2.25) |
| Non-persistently infected | 78,902 | 4,291 (5.4) | 100 (91–115) | Reference | Reference |
| *Long COVID at first assessment 26 weeks or longer post-infection* | | | | | |
| Persistently infected | 326 | 19 (5.8) | 312 (271–390) | 1.44 (0.90–2.29) | 1.24 (0.77–2.00) |
| Non-persistently infected | 72,608 | 3,000 (4.1) | 320 (272–384) | Reference | Reference |

CI, confidence interval; IQR, interquartile range; OR, odds ratio.

they were still experiencing symptoms more than 4 weeks after they first had COVID-19 (see Methods). We estimated the prevalence of self-reported long COVID in individuals with persistent infection compared with individuals with non-persistent infection, accounting for several confounding variables (see Methods). In the persistent infection group, 9.0% of respondents (32 of 356) self-reported long COVID at their first visit 12 weeks or longer since the start of infection, and 5.8% (19 of 326) reported long COVID at 26 weeks or longer. However, among the non-persistently infected group, only 5.4% (4,291 of 78,902) reported long COVID at their first visit 12 weeks or longer, and 4.1% (3,000 of 72,608) reported long COVID at 26 weeks or longer.

Correcting for confounders, we found strong evidence for a 55% higher odds of reporting long COVID at 12 weeks or more post-infection among individuals with persistent infection than individuals with non-persistent infection ($P = 0.004$ for the unadjusted model; $P = 0.021$ for the adjusted model), but no evidence of a difference for long COVID at 26 weeks or more post-infection ($P = 0.127$ for the unadjusted model; $P = 0.367$ for the adjusted model) (Table 3). The lower probability of reporting long COVID 26 weeks post-infection than at 12 weeks post-infection could be because the majority of the persistent infections that we identified lasted for less than 3 months, and hence persistence of an infection may no longer be a contributing factor to long COVID beyond 3 months.

## Discussion

We developed a robust approach for identifying persistent SARS-CoV-2 RNA in individuals with sequenced samples spanning 1 month or longer. Evidence suggests that these represent persistent infections; however, persistence of non-replicating viral RNA cannot be categorically ruled out in all cases. Because viral genetic data are needed to confirm persistent infection, we can only identify persistent infections in individuals with at least two high viral load (Ct ≤ 30) samples. Given this, the number of persistent infections that we identified should be considered a lower bound. Of the 381 persistent infections that we identified among participants of the ONS-CIS, 54 lasted at least 2 months and two over 6 months; in some cases, the infecting lineage had gone extinct in the general population. By contrast, we only identified 60 reinfections by the same major lineage as the primary infection, suggesting that immunity to the same variant remains strong after infection, at least until the lineage has gone extinct (Table 1).

The large number of persistent infections that we uncovered is striking, given the leading hypothesis that many of the variants of concern emerged wholly or partially during long-term chronic infections in immunocompromised individuals[1]. As the ONS-CIS is a community-based surveillance study, our observations suggest that the pool of people in which long-term infections could occur, and hence potential sources of divergent variants, may be much larger than generally thought. However, we do not know whether the individuals with persistent infection that we identified have other health conditions

that may make them more susceptible to these long infections. We estimate that 1 in 1,000 of all infections, and potentially as many as 1 in 200, may become persistent, with intermittent high viral loads, for at least 2 months.

Our results are consistent with a household study[37] in which 6% of infections (7 of 109) have been reported to have viral shedding after 30 days since the onset of symptoms, but only two had a Ct ≤ 30 after 25 days, and none after 30 days. By contrast, a study of hospitalized individuals[38] has reported prolonged shedding in 18% (17 of 92) of patients. This much higher rate than individuals sampled in the community regardless of symptoms, as in our study, probably represents the severity of infection among the hospitalized individuals.

The harbouring of persistent infections in the general community may also help to explain the early detection of cryptic lineages circulating in wastewaters[39,40] long before they spread in the population at large. In support of the hypothesis that variants of concern may emerge during prolonged infections, several studies have shown elevated evolutionary rates driven by selection during chronic infections of immunocompromised individuals[6–8]. Among many of the individuals with persistent infection that we identified, we observed long periods of evolutionary stasis at the consensus level, indicating little to no directional selection during infection. In HIV, zero synonymous consensus differences between sequences spanning prolonged periods of within-host infection have also been observed, probably because synonymous mutations are under little or no selective pressure[37,38]. However, in other persistent infections, we found strong evidence for positive selection and parallel evolution, particularly in *S* and *ORF1ab*. In the most extreme cases, we observed one persistent infection with zero consensus change for over 150 days, whereas another persistent infection had 33 substitutions over a 4-month period, 20 of which were non-synonymous, and where the great majority of these mutations emerged during the first 30 days after the first positive sequence.

Most of the persistent infections in our study with at least three positive PCR samples over the course of infection showed a pattern of viral rebound (high to low to high viral load). This suggests that the mechanism of persistent infection is not due to delayed clearance of the virus by the host, but points to possible presence of actively replicating virus. Other studies have also reported viral rebound both during acute[41] and chronic[42] infections. These rebounding dynamics also exacerbate the difficulty of distinguishing between persistent and reinfections in the absence of sequence data. A common criterion for identifying reinfections is to only consider positive PCR samples that are at least 90 days apart[43]. An advantage of the genetic approach used in our study is not only the ability to detect reinfections over shorter timescales of less than 60 days but also to rule out reinfection over longer timescales (more than 90 days). Our findings are in broad agreement with recent systematic reviews showing lower rates of reinfection during the first 12 weeks since the initial infection[44].

Individuals with persistent infections report fewer symptoms later in a persistent infection than at their first positive sample, or remain

asymptomatic throughout infection, but have more than 50% higher odds of long COVID than a group of individuals with non-persistent infection. Although the link between viral persistence and long COVID may not be causal, these results suggest that persistent infections could be contributing to the pathophysiology of long COVID[10,45], as also evidenced by the observation of circulating SARS-CoV-2 S1 spike protein in a subset of patients with long COVID months after first infection[46]. There is also a growing body of evidence of the persistence of replication-competent virus throughout the body months after the start of an infection[47,48], and very recently that this persistence is strongly associated with higher risk of long COVID[49].

The association between persistent infection and long COVID does not imply that every persistent infection can lead to long COVID (only 9% of individuals with persistent infection reported having long COVID) nor does it mean that all cases of long COVID are due to a persistent infection. Indeed, many other possible mechanisms have been suggested to contribute to long COVID, including autoimmunity/inflammation, organ damage, Epstein–Barr virus reactivation and microthrombosis (see ref. 10 for a recent review).

Together, our observations highlight the continuing importance of community-based genomic surveillance both to monitor the emergence and spread of new variants, and to gain a fundamental understanding of the natural history and evolution of novel pathogens and their clinical implications for patients.

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

## Methods

### ONS-CIS

This work contains statistical data from ONS, which is Crown Copyright. The use of the ONS statistical data in this work does not imply the endorsement of the ONS in relation to the interpretation or analysis of the statistical data. This work uses research datasets that may not exactly reproduce National Statistics aggregates.

The ONS-CIS is a UK household-based surveillance study in which participant households are approached at random from address lists across the country to provide a representative sample of the population[21]. All versions of the study protocol are available at https://www.ndm.ox.ac.uk/covid-19/covid-19-infection-survey/protocol-and-information-sheets. All individuals 2 years of age and older from each household who provide written informed consent provide swab samples (taken by the participant or parent or carer for those under 12 years of age), regardless of symptoms, and complete a questionnaire at assessments. The survey offered participants the option of only having one enrolment assessment (taken by approximately 1%), or weekly assessments for only 1 month (taken by approximately 1%; Extended Data Fig. 1). All other enrolled participants (approximately 98%) were assessed weekly for the first month of their enrolment in the survey and then approximately monthly (originally for 1 year; all such participants were approached for re-consent for ongoing follow-up beyond 1 year). The survey had rolling recruitment to meet its target for taking a certain number of swabs from the population each month, but in practice, most recruitments occurred between September and December 2020 (Supplementary Information; also see supplementary table 4 in ref. 50). The rolling recruitment enabled the study to achieve its overall sample numbers (required to address its surveillance objectives) while accounting for participants withdrawing from the study. As is standard, the protocol also allowed a 14-day window around the approximately monthly assessments (shifting any following assessments to avoid swabbing participants again at very short (and variable) notice); crucially, assessments were not missed to meet survey targets.

As the vast majority of recruitment comes from invitations sent to households randomly selected from address lists that we do not have relevant demographic information, we are not able to compare characteristics of those agreeing and not agreeing to participate. From 26 April 2020 to 31 July 2022, assessments were conducted by study workers visiting each household; from 14 July 2022 onwards, assessments were remote, with swabs taken using kits posted to participants and returned by post or courier, and questionnaires completed online or by telephone. For this analysis, we included data from 2 November 2020 to 15 August 2022, spanning a period from Alpha to Omicron BA.2 sequences within the ONS-CIS dataset (Extended Data Table 1).

To date, of 535,731 participants recruited into the ONS-CIS, 109,417 (20%) have either completed their participation after a single enrolment visit, visits only for the first month or only for the first year (7%) or withdrawn (13%; see Supplementary Information). Moving house was a major reason for completing participation in the survey (as this leads to participants no longer being eligible for follow-up as it is the original address that is sampled), a small number of participants died (0.4%), and in July 2022, the survey moved to a remote data collection approach at which point some participants chose to end their participation. For the time period of this study, 96.2% of swabs had a negative result and 1.9% had a positive result (1.9% were void). For those with positive test results, the mean time since the previous assessment was 35.2 days and to the next assessment was 37.1 days. For those with a negative test, the associated numbers were 31.8 days and 33.0 days. By definition, 100% of first positive samples from each persistent infection had a subsequent assessment. There was no statistical difference in the time between sampling for individuals with persistent infection compared with those testing positive (Supplementary Information).

### Sequencing

From December 2020 onwards, sequencing was attempted on all positive samples with Ct ≤ 30; before this date, sequencing was attempted in real time wherever possible, with some additional retrospective sequencing of stored samples. The vast majority of samples were sequenced on Illumina Novaseq, with a small number using Oxford Nanopore GridION or MINION. One of two protocols were used: the ARTIC amplicon protocol[51] with consensus FASTA sequence files generated using the ARTIC nextflow processing pipeline (v1)[52], or veSeq, an RNA sequencing protocol based on a quantitative targeted enrichment strategy[19,53] with consensus sequences produced using shiver (v1.5.8)[54]. During our study period, we identified 94,943 individuals with a single sequence and 5,774 individuals with two or more sequences. Here we only included sequences with 50% or more genome coverage.

### Identifying candidate persistent infections

We first identified individuals with two or more sequenced samples taken at least 26 days apart. We chose this cut-off because the majority of individuals with acute infection shed the virus for less than 20 days and no longer than 30 days in the respiratory tract[24,55]. Given the extreme heterogeneity in the shedding profiles during some acute infections[24,55], we also considered a more conservative 56-day cut-off for some analyses. Selection was based on availability of sequences, which were required for genetic analysis; it was not possible to allow for failure to identify any long-term shedding due to participants not having assessments/swabs or tests failing or subsequent positives having Ct > 30, and therefore not being sent for sequencing. However, this means that some persistent infections are likely to have been missed and so our estimates should be considered a lower bound.

Candidate persistent infections were defined in one of two ways: (1) pairs of sequenced samples that belonged to the same major lineage, and (2) pairs of sequenced samples where one or both had no defined phylogenetic lineage, but where the genetic distance between them was lower than that required to differentiate two major lineages (Extended Data Fig. 9). The major lineages that we considered were Alpha (B.1.1.7), Delta (B.1.617.2), Omicron BA.1 and Omicron BA.2, including their sublineages. We assumed pairs belonging to different major lineages were either co-infections or reinfections with two different virus lineages. Only candidate persistent infections were considered in further analysis.

### Identifying persistent infections

We determined whether two sequences from the same individual are from the same infection by whether they share a rare SNP at two or more consecutive time points relative to the population-level consensus. If an intermediate sequence from that individual had an unknown nucleotide at a site (due to poor coverage), whereas the first and last sequences shared a rare SNP, then the intermediate sequence was also assumed to be part of the same infection. Rare SNPs were defined as those that were shared by fewer than a threshold number of sequences, belonging to each major lineage, within the full ONS-CIS dataset (Extended Data Fig. 2). The thresholds were chosen to maximize the number of persistent infections identified while minimizing the number of false positives (see below).

To determine the false-positive rate, for each major lineage, we generated a dataset of 1,000 randomly paired sequences from different individuals in the ONS-CIS, each sampled at least 26 days apart. We determined the proportion of these pairs that would have been incorrectly identified as persistent infections as a function of the threshold for determining whether a SNP is rare (Extended Data Fig. 2). Although the total number of persistent infections that we identified (among the list of candidate persistent infections) grew as the threshold for determining whether a SNP is rare increased, at very high thresholds, the rate of false positives (among the list of randomly paired sequences)

was also high. In our study, we chose a threshold of 400 sequences (corresponding to all sequences of the same major lineage within the full ONS-CIS dataset) for all of the major lineages, giving a false-positive rate (identifying an infection as persistent when it was not) of 0–3%. Using this threshold, approximately 92–98% of all sequences from the four major lineages had a rare SNP relative to the major-lineage population-level consensus.

### Identifying reinfections with the same major lineage

Any pair of sequences from the same individual, of the same major lineage and at least 26 days apart were considered as candidate reinfections. Of these, pairs that had at least one nucleotide difference at the consensus level, and did not share any rare SNPs, were classed as reinfections. Pairs that had no identical rare SNPs, nor any nucleotide differences at the consensus level, were classed as undetermined.

Sample mix-ups could inflate the true number of reinfections. In the ONS-CIS, each sample has a unique barcode, a small minority of barcodes are positive, and even fewer still have a Ct ≤ 30; therefore, random swapping of barcodes is unlikely to result in a wrong positive sample with Ct ≤ 30 being sent for sequencing. For each weekly sampling batch, we also checked concordance between lineage from the sequencing laboratory and S gene target failure from the testing laboratory; concordance between Ct from the testing laboratory and genome coverage from the sequencing laboratory (high coverage is expected for low Ct, and low coverage for high Ct); and for veSeq, a log-linear relationship between the number of mapped reads from the sequencing laboratory and Ct from the testing laboratory[19].

### Phylogenetic analysis

For each of the four major lineages, we chose 600 consensus sequences with at least 95% coverage from the ONS-CIS dataset using weighted random sampling, with each sample of major lineage $i$ collected in week $j$ given a weight $1/n_{ij}$, where $n_{ij}$ is the number of sequences of major lineage $i$ collected during week $j$[22]. These sequences were added as a background set to the collection of all consensus sequences for samples from persistent infections and reinfections. Mapping of each sequence to the Wuhan-Hu-1 reference sequence was already performed by shiver, and thus a full alignment for each of the four lineages could be constructed using only this.

Maximum likelihood phylogenetic trees were constructed using IQ-TREE (v1.6.12)[56] using the GTR+gamma substitution model and ultrafast bootstrap[57]. Each tree was rooted using the collection dates of the samples and the heuristic residual mean square algorithm in TempEst[58]. Visualization used ggtree[59].

### Measuring the number of independent appearances of mutations and their fitness effects

To find the frequency with which mutations (not including deletions) that we identified during persistent infections are represented in cross-sectional samples from the population and their between-host level fitness, we used the results from ref. 29 on the estimated number of appearances of mutations from a representative global dataset of approximately 6.5 million SARS-CoV-2 sequences (for number of appearances: https://github.com/jbloomlab/SARS2-mut-fitness/blob/main/results/mutation_counts/aggregated.csv; for estimating the fitness effect of mutations: https://github.com/jbloomlab/SARS2-mut-fitness/blob/main/results/aa_fitness/aamut_fitness_by_clade.csv), as well as a subset of those sequences that are only sampled from England (arguably more relevant to our sequences from the ONS-CIS). When doing this, we controlled for major lineage, meaning, for example, if a mutation occurred in a BA.1 persistent infection, we only considered the number of times it appeared on the BA.1 phylogeny. To map between Pangolin lineages and Nextstrain clades, we assumed B.1.1.7 ≡ 20I, B.1.617.2 ≡ {21A,21I,21J}, BA.1 ≡ 21K and BA.2 ≡ {21L,22C,22D}. We also compared the frequency and fitness effect of mutations that appeared in two persistent infections (that is, recurrent mutations) and those that appeared in only one persistent infection (that is, single mutations) as reported in ref. 29.

### Estimating the percentage of infections that are persistent

We identified 381 and 54 infections that lasted 30 days or longer and 60 days or longer, respectively. Comparing this with the number of individuals that had sequenced samples belonging to Alpha, Delta, BA.1 or BA.2, we identified approximately 0.49% (381 of 77,561) and 0.07% (54 of 77,561) of infections with at least one sample that could be sequenced as persistent for 30 days or longer and 60 days or longer, respectively. As the ONS-CIS is a representative sample of individuals from the general population, we can estimate the percentage of all SARS-CoV-2 infections that became persistent for 1 month or longer, and that have intermittent high viral loads. To do this, we need to determine the probability that a persistent infection with one sequenced sample has at least one more sequenced sample. As most persistent infections probably last 1–3 months, and without knowing the true viral kinetics during persistent infection, this can be approximated as the probability that a persistent infection has virus that can be sequenced on any given day of sampling.

At one extreme, if a typical persistent infection has a virus sample that can be sequenced for only 4 days per month (assuming viral dynamics similar to one acute infection each month), only 14% of persistent infections would be detected through approximately monthly sampling. Correcting for this, we would estimate the percentage of detected infections that are persistent in the general population for 30 days or longer to be 3.5%, calculated as the ratio of the estimated prevalence of persistent infections (0.49%) to the detection rate (14%). Similarly, for infections persisting 60 days or longer, the estimated percentage would be 0.5% (0.07%/0.14). At the other extreme, if we assume typical persistent infections have sequenceable virus for 20 days per month and, therefore a detection rate of 71%, we would estimate the percentage of detected infections that are persistent infections in the general population for 30 days or longer to be 0.7% (0.49%/0.71) and for 60 days or longer to be 0.1% (0.07%/0.71).

### Comparing viral load activities and symptoms

To quantify the changes in viral load activities during persistent infections, we compared Ct values at the last time point a sequence was obtained to when the first sequence was collected. Likewise, for reinfections, we compared the changes in Ct value between the primary infection and reinfection. We used a paired Student's $t$-test to calculate $P$ values in both cases as the distribution of differences in Ct values were normally distributed for both persistent infections ($W = 0.99$, $P = 0.28$) and reinfections ($W = 0.99$, $P = 0.78$) as determined by the Shapiro–Wilk test[60].

We also tracked 12 symptoms consistently solicited from all participants at every assessment. Symptoms were fever, weakness/tiredness, diarrhoea, shortness of breath, headache, nausea/vomiting, sore throat, muscle ache, abdominal pain, cough, loss of smell and loss of taste. At each follow-up assessment, participants were asked whether these 12 symptoms had been present in the past 7 days (mandatory question completed at all assessments where a swab was taken). Symptom discontinuation was defined as the first occurrence of two successive follow-up visits without reporting symptoms. To compare symptom counts during persistent infections and reinfections, we used the two-sided paired Wilcoxon test as the distribution of symptom differences is not normally distributed (Fig. 3e). For calculation of $P$ values and visualization of histograms and box plots, we used Mathematica (v13.1.0.0).

### Long COVID analysis

Attributing persistent symptoms to a previous SARS-CoV-2 infection is difficult in the absence of a diagnostic test for long COVID,

and long COVID cases are known to be under-recorded in electronic health records[61]. Long COVID status was therefore self-reported by study participants, so we cannot exclude some participants' symptoms being caused by a medical condition other than COVID-19. From February 2021, at every assessment, participants were asked "would you describe yourself as having long COVID, that is, you are still experiencing symptoms more than 4 weeks after you first had COVID-19, that are not explained by something else?".

When estimating long COVID prevalence in this analysis, we considered the first assessment at least 12 weeks and at least 26 weeks after infection. Our comparison group comprised all individuals with a positive PCR test and Ct ≤ 30 at the first positive test, excluding the individuals with persistent infection identified in this study, over the same time span as persistent infections such that first positive test was within the range of dates of the first positive test among the persistent infection group. Although the underlying study design for ONS-CIS is a cohort study, this specific analysis of long COVID focuses on comparing persistent to non-persistent infections in terms of the risk of subsequent self-reported long COVID (binary outcomes, at least 12 weeks and at least 26 weeks following the first positive test). Some missing data were inevitable, given the timeframe of the study and participant completion or withdrawal (see above); overall, the long COVID question was not completed at 368,161 of 6,797,789 (5.4%) of assessments during the study period from 4 February 2021 when it was introduced, with 93% and 86% of participants without persistent infection but with a positive test with Ct < 30 having a response to the long COVID question at least 12 and 26 weeks after infection, respectively (Extended Data Fig. 1). Analysis used complete cases, that is, excluded those who did not have a response to the long COVID question in this timeframe (Extended Data Fig. 1). As these are binary outcomes rather than a time-to-event outcome, either an odds ratio or a relative risk could be used to evaluate the risk of long COVID in individuals with persistent infection; here we used odds ratio. The fact that some persistent infections were probably missed due to sequencing only being attempted in high viral load samples and due to missed assessments means that our estimates of the impact of persistent infection are likely to be biased towards the null, that is, the true effects of persistent infection are probably larger than we estimate. Follow-up from the start of infection to first long COVID response was similar between persistent and non-persistent infections (Table 3).

In calculating the odds ratio of long COVID in individuals with persistent infection relative to the comparison group, we used a binary logistic regression model and accounted for confounding variables such as age at the last birthday, sex, Ct value, calendar date, area deprivation quintile group, presence of self-reported long-term health conditions (binary), vaccination status (unvaccinated or single vaccinated, fully vaccinated or booster vaccinated 14–89 days ago, fully vaccinated or booster vaccinated 90–179 days ago, fully vaccinated or booster vaccinated 180 or more days ago) and days from first positive test to long COVID follow-up response. All variables except the last one were defined at the time of the first positive test. Continuous variables (age, Ct value, calendar date and days to follow-up response) were modelled as restricted cubic splines with a single internal knot at the median of the distribution and boundary knots at the 5th and 95th percentiles. Vaccination status was derived from a combination of CIS and National Immunisation Management System (NIMS) data for participants in England, and CIS data alone for participants in Wales, Scotland and Northern Ireland. Given the number of potential confounders included, we did not test for interaction (effect modification). We did not test for goodness of fit because the model was solely used to control for measured confounders of the relationship between persistent positivity and long COVID, which we selected on substantive, rather than empirical, grounds (that is, using a causal inference approach).

Although we controlled for many confounders that could potentially impact our long COVID analysis, of note, age, sex, vaccination status

and previous infection, there may still be unknown residual confounders that can influence our results. We were also unable to perform the long COVID analysis for the reinfection group due to the low number of participants in this cohort who reported new-onset long COVID 12 weeks or longer or 26 weeks or longer after infections.

## Reporting summary

Further information on research design is available in the Nature Portfolio Reporting Summary linked to this article.

## Data availability

All raw consensus sequences have been made publicly available as part of the COG-UK Consortium[62] (https://webarchive.nationalarchives. gov.uk/ukgwa/20230505214946/https://www.cogconsortium.uk/ priority-areas/data-linkage-analysis/) and are available from the European Nucleotide Archive at EMBL-EBI under accession number PRJEB37886. These sequences can be accessed using their COG-UK sample title. All post-aligned consensus sequences (aligned to Wuhan-Hu-1 reference sequence) are available on figshare to facilitate reproducibility of our findings (https://figshare.com/s/acdaf46f87e0f9874e38). All remaining data, excluding personal clinical information on participants, are available in the main text and supporting materials. Source data are provided with this paper.

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

**Acknowledgements** We thank R. Neher and J. Bloom for their constructive feedback on this work. The CIS was funded by the Department of Health and Social Care and the UK Health Security Agency, with in-kind support from the Welsh Government, the Department of Health on behalf of the Northern Ireland Government and the Scottish Government. The COVID-19 Infection Survey Group of the COVID-19 Genomics UK (COG-UK) Consortium was supported by funding from the Medical Research Council part of UK Research & Innovation, the National Institute of Health Research (NIHR) (grant code: MC_PC_19027) and Genome Research Limited, operating as the Wellcome Sanger Institute. We acknowledge use of data generated through the COVID-19 Genomics Programme funded by the Department of Health and Social Care. A.S.W. is supported by the NIHR Health Protection Research Unit in Healthcare Associated Infections and Antimicrobial Resistance at the University of Oxford in partnership with the UK Health Security Agency (NIHR200915) and the NIHR Oxford Biomedical Research Centre, and is an NIHR Senior Investigator. T.H. is supported by the Royal Society and Alan Turing Institute for Data Science and Artificial Intelligence. K.L. is supported by the Royal Society and the Wellcome Trust (107652/Z/15/Z) and by the Li Ka Shing Foundation. The research was supported by the Wellcome Trust Core Award grant number 203141/Z/16/Z, with funding from the NIHR Oxford BRC. The views expressed are those of the authors and not necessarily those of the NHS, the NIHR, the Department of Health, the Department of Health and Social Care or the UK Health Security Agency.

**Author contributions** M.G. and K.L. wrote the original draft of the manuscript. M.G., M.H., T.G., D.A., H.R.F., L.F., M.B., A.S.W. and K.L. reviewed and edited the manuscript. M.G., M.H., T.G., D.A., T.H., H.R.F., A. Nurtay, L.F., R.S., E.R., R.C., A.S.W. and K.L. performed the analysis. G.M.-C., A.G., A.T., P.P., L.J.L., D.L.S., A. Nelson, M.C., C.M., G.R.Y., R.A.N.d.S., Z.R., M.A.T., the Wellcome Sanger Institute COVID-19 Surveillance Team, the COVID-19 Infection Survey Group, the COG-UK Consortium and D. Bonsall generated the data. L.T., S.A.K., D.L.S., A. Nelson, M.C., C.M., G.R.Y., R.A.N.d.S., Z.R., M.A.T., the Wellcome Sanger Institute COVID-19 Surveillance Team and D. Bonsall processed the data. A.S.W. and K.L. came up with the study design. D. Buck, J.B., C.F., D. Bonsall, A.S.W. and K.L. supervised sequencing and data processing. K.L. supervised the analysis.

**Competing interests** The authors declare no competing interests.

**Additional information**
**Correspondence and requests for materials** should be addressed to Mahan Ghafari or Katrina Lythgoe.

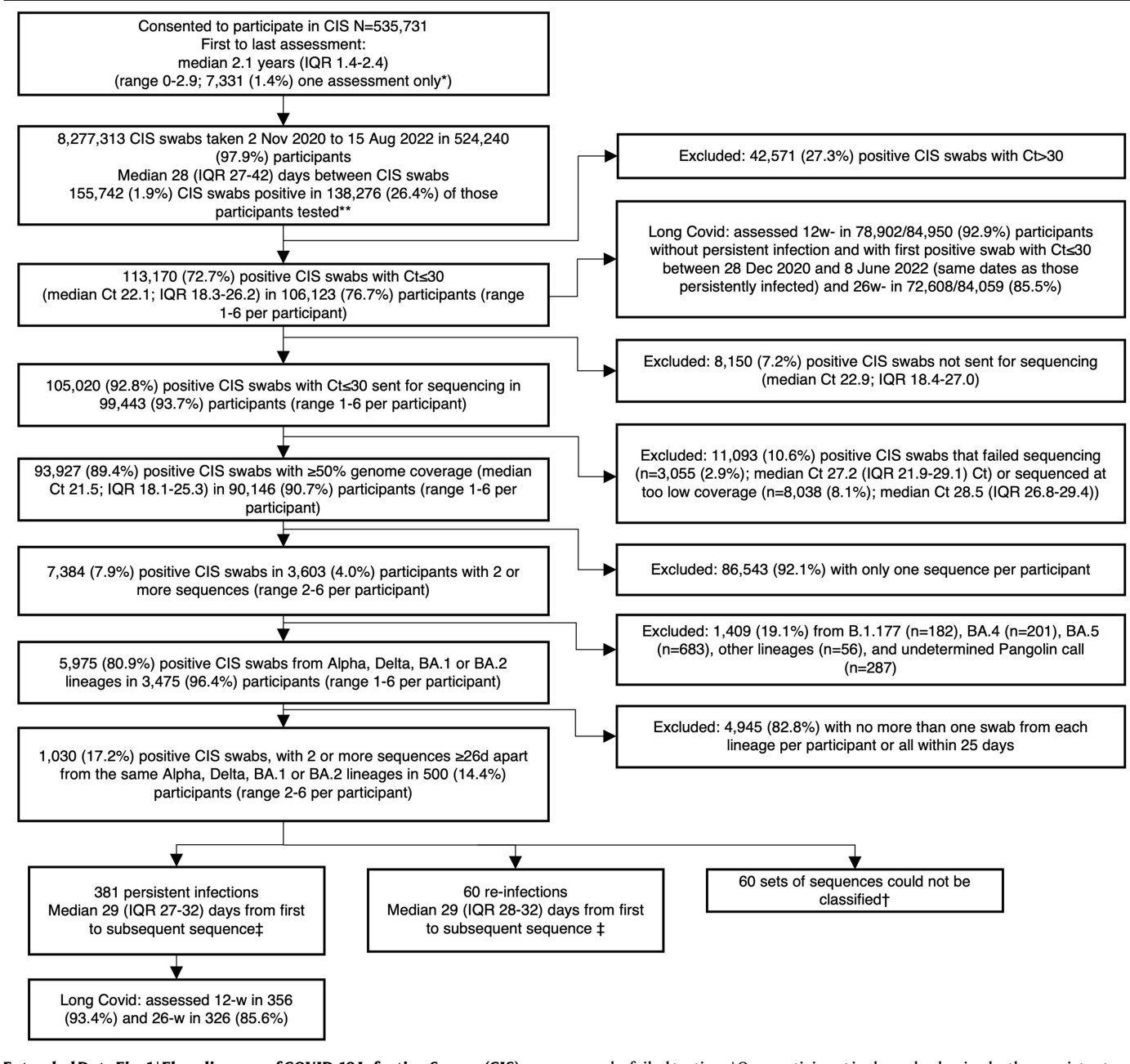

**Extended Data Fig. 1 | Flow diagram of COVID-19 Infection Survey (CIS) participant in this study.** *At enrolment, participants could choose to have one assessment only, or 5 assessments over the first month only, or to continue approximately monthly follow-up until the end of the study. **158,719 (1.9%) CIS swabs failed testing. †One participant is classed as having both a persistent infection and reinfection with the same major lineage. ‡ Two-sided ranksum p = 0.53.

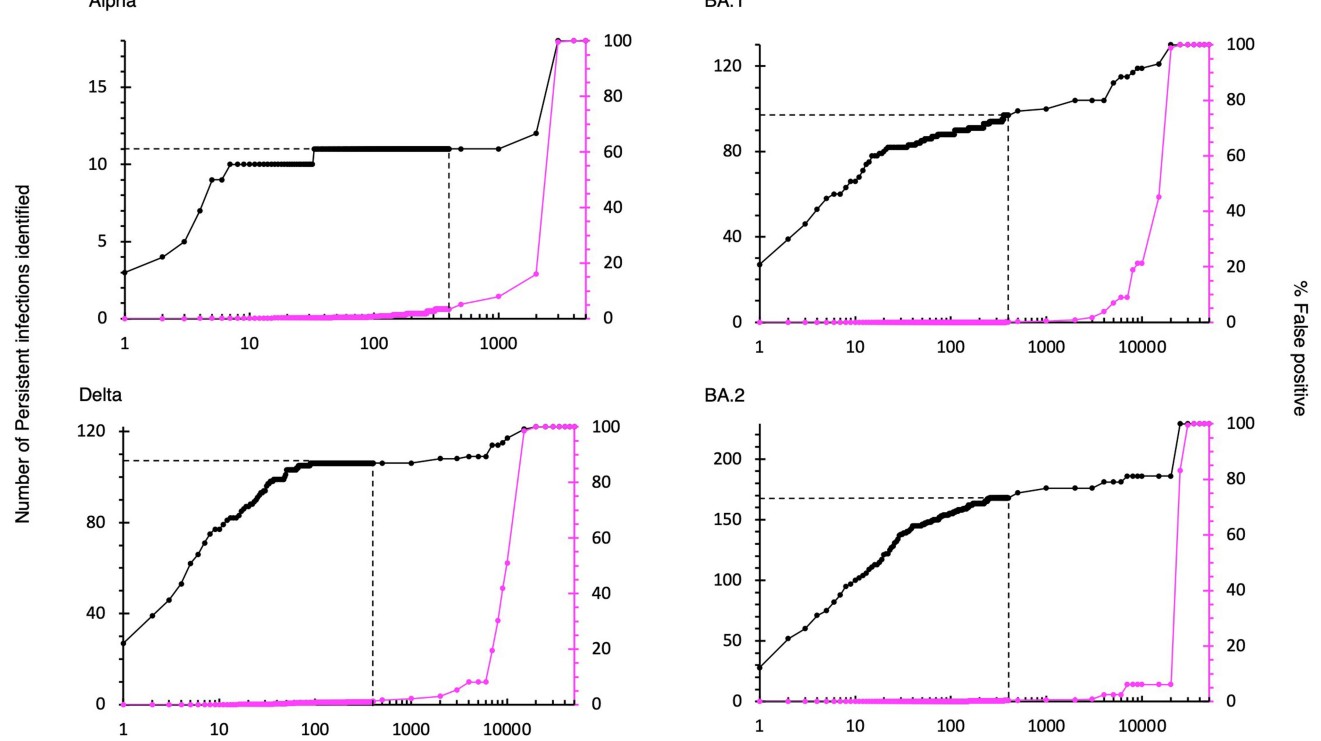

Threshold number for calling a shared rare SNP

**Extended Data Fig. 2 | Number of persistent infections identified with a shared rare SNP as a function of the threshold number of cases for calling a rare SNP.** A threshold value of 1 for a rare SNP means the rare SNP is only found in one sequence of that lineage in the ONS-CIS dataset, excluding sequences from any persistently infected individuals. The number of persistent infections identified gives the number of persistent infections lasting at least 26 days we would identify as persistent in the ONS-CIS using the given threshold (black). The false positive percentage gives the percentage of times two random samples of the same major lineage taken from the ONS-CIS would be falsely identified as belonging to the same persistent infection (magenta; 1,000 pairs of samples were considered). As the threshold value for calling a rare SNP

increases, the number of persistent infections identified (black) increases, but so does the false positive rate. We chose a threshold number of 400 (vertical dashed line) in this study for identifying persistent infections, since for this threshold the percentage of false positives were 0% for BA.1 and BA.2 and 3% for Alpha and Delta, but the number of persistent infections identified has begun to plateau. The total number of candidate persistent infections (that have at least a pair of sequence that are ≥26 days apart) we considered for each lineage equals the number of infections identified when there is a false positive rate of 100% (18 Alpha, 122 Delta, 130 BA.1, and 230 BA.2). The exception is a single individual with two BA.2 sequences which do not have a shared SNP relative to the BA.2 population-level consensus.

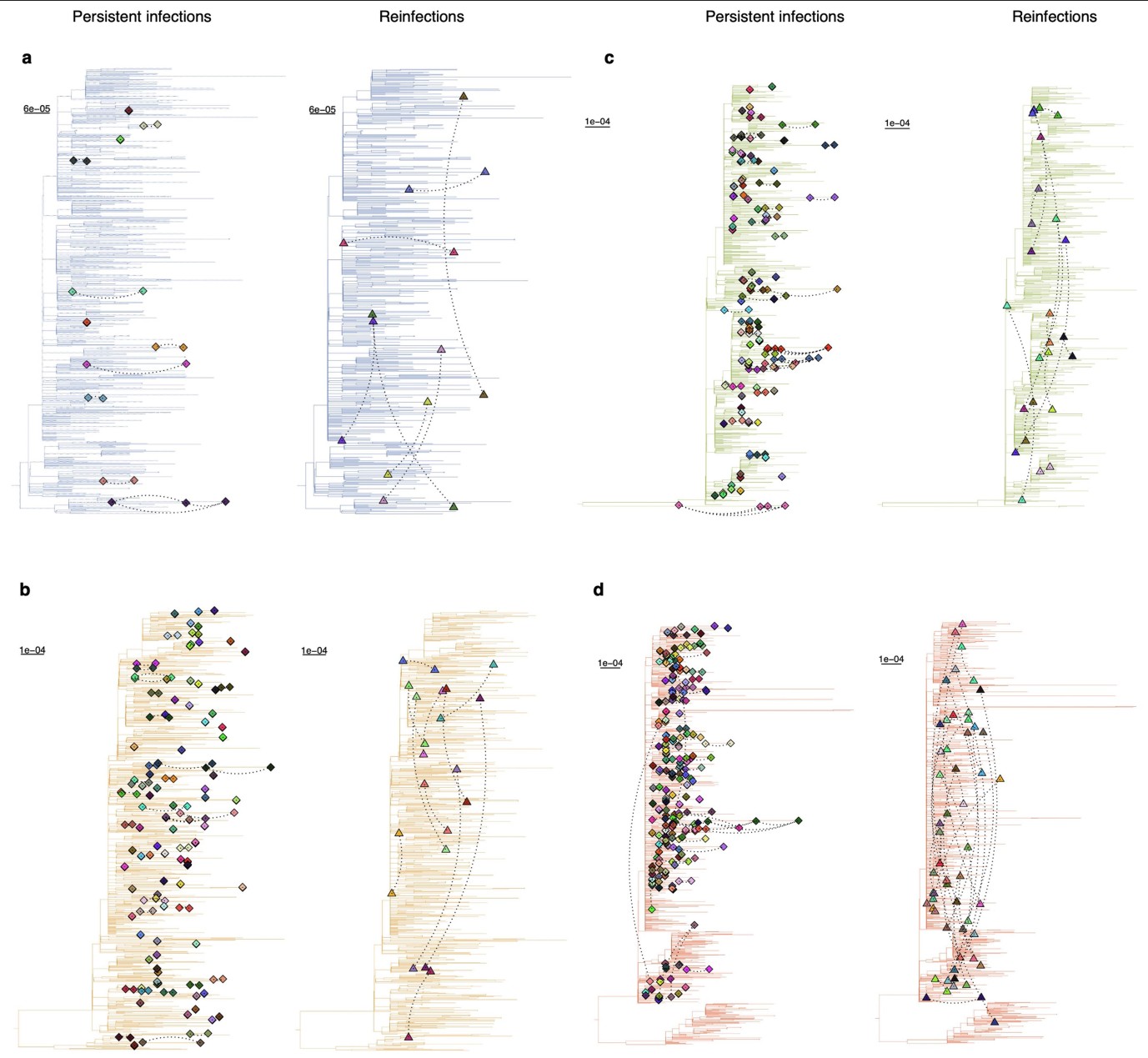

**Extended Data Fig. 3 | Phylogenetic relationship between samples from persistent infections and a representative background population per major lineage.** Dashed lines connect every pair of sequences from the same individual. All sequences from the same individual are given the same colour. Pairs of sequences for (**a**) Alpha, (**b**) Delta, (**c**) Omicron BA.1, and (**d**) Omicron BA.2 that belong to persistent infections cluster closely together while reinfections do not. However, some of the sequences in 2 (out of 97) persistent infections with BA.1 and 5 (out of 167) persistent infections with BA.2 have poor bootstrap support (<80) and do not cluster together or cluster in a basal sister relationship. In all of these 7 cases, at least one of the sequences from each individual had a Ct value close to 30 with poor coverage. On the other hand, all sequences that belong to the same individual and have strong bootstrap support (>80) cluster together.

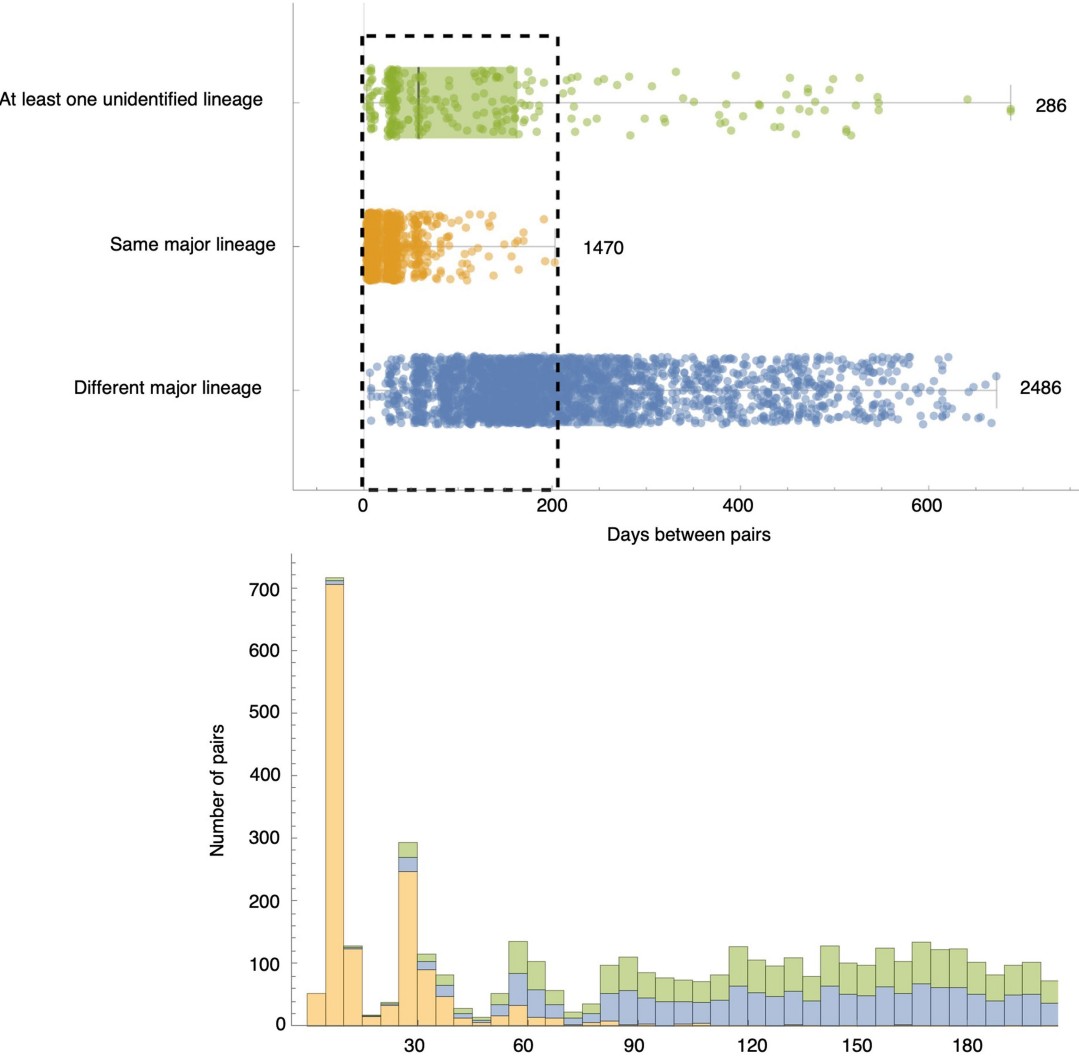

**Extended Data Fig. 4 | Days between all pairs of sequences from the same individual with two or more sequences.** Pairs of sequences are classified as (i) pairs with at least one unidentified Pango lineage (green), (ii) pairs with identical major lineage (orange), and (iii) pairs from different major lineages (blue). The boxes indicate the interquartile range (IQR), which spans from the 25th to the 75th percentile, with the centre being the median and marked by a black vertical line. The medians for categories (i), (ii), and (iii) are 58, 9, and 180 days, with IQRs of 28–163 days, 7–28 days, and 123–280 days, respectively. The extremities (displayed as grey horizontal lines) denote the minimum and maximum values within each category. Bottom panel shows the counts of pairs in each of these three categories for the first 200-day time span (highlighted in a dashed rectangle in the top panel). Pairs include all possible combinations of sequences from the same individual, including sequences that are less than 26 days apart from each other. The number of pairs peaks at the 7-, 30-, and 60-day periods due to the sampling frequency of ONS-CIS (see Methods). Note that pairs with identical major lineage may not necessarily have identical Pango lineages (see Methods).

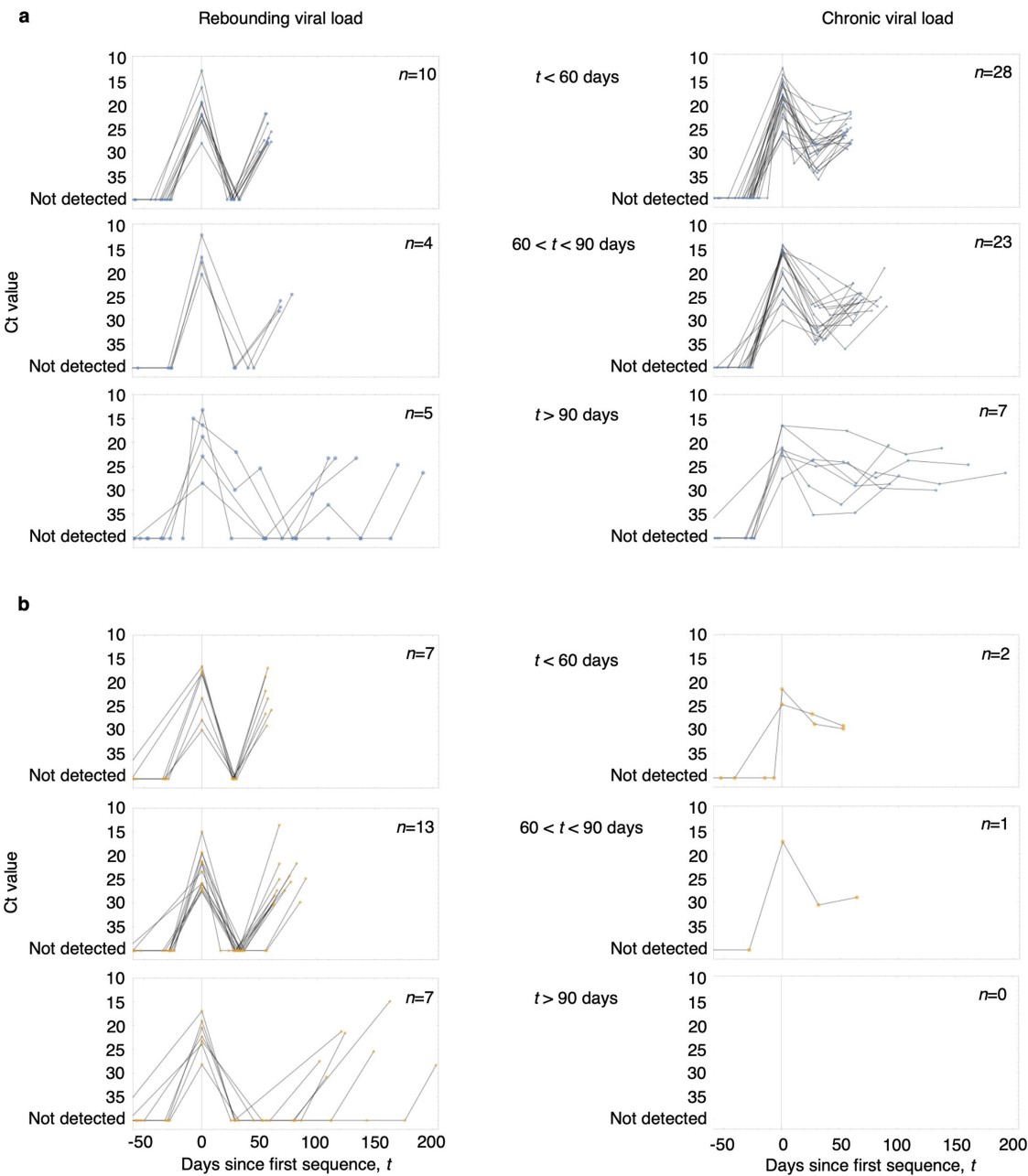

**Extended Data Fig. 5 | RNA viral load dynamics of individuals identified with persistent infections and reinfections stratified by duration and viral activity.** RNA viral load activities of individuals, with 3 or more PCR tests taken during infection/until reinfection, identified as having (**a**) persistent infections and (**b**) reinfections with rebounding (i.e., a negative RT-PCR test during the infection) (left column) and persistent chronic (right column) trajectories. Three reinfections (two occurring in <60 days and one between 60 to 90 days since first sequence) with persistent chronic viral load dynamics are excluded from the reinfection group as they are deemed potential persistent infections which do not have rare SNPs.

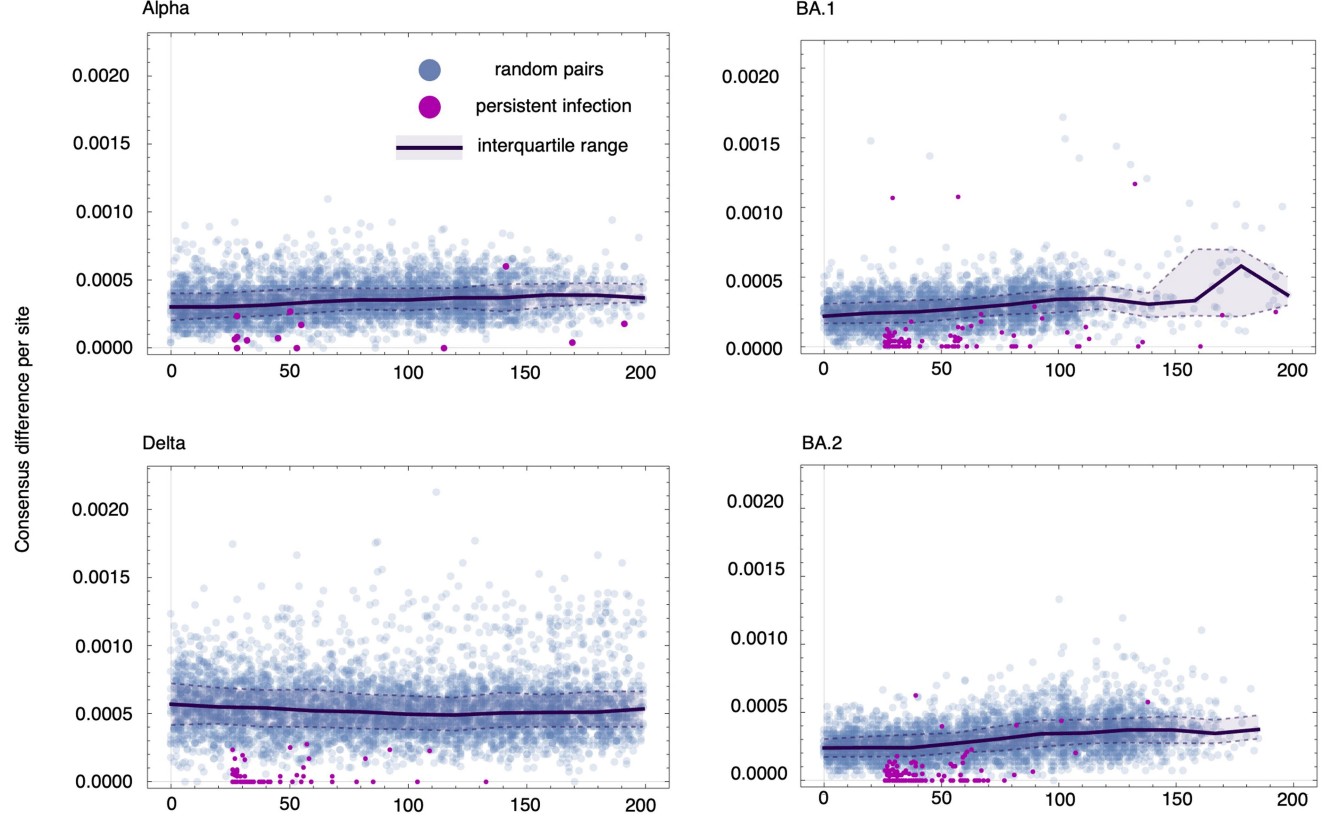

**Extended Data Fig. 6 | Number of single nucleotide polymorphisms detected in pairs of sequences from persistent infections vs. random pairs from a representative background population.** Number of consensus nucleotide differences per site between all the sequences collected from persistent infections (purple) and random pairs from individuals with only a single sequence within the ONS-CIS (blue) as a function of the number of days between each pair. For each major lineage, a pool of sequences from individuals with only one sequence within the ONS-CIS was sub-sampled and 500 random pairs generated for every 20 additional days between samples. For some major lineages where there were fewer than 500 pairs available beyond a certain time point, all possible random pairs within that 20-day period are used. Solid line and shaded area show the median and interquartile range, respectively, for random pairs over time. Note that the line and shaded area in each graph does not represent the rate of evolution but can be deemed as a measure of lineage diversity as a function of time difference between samples.

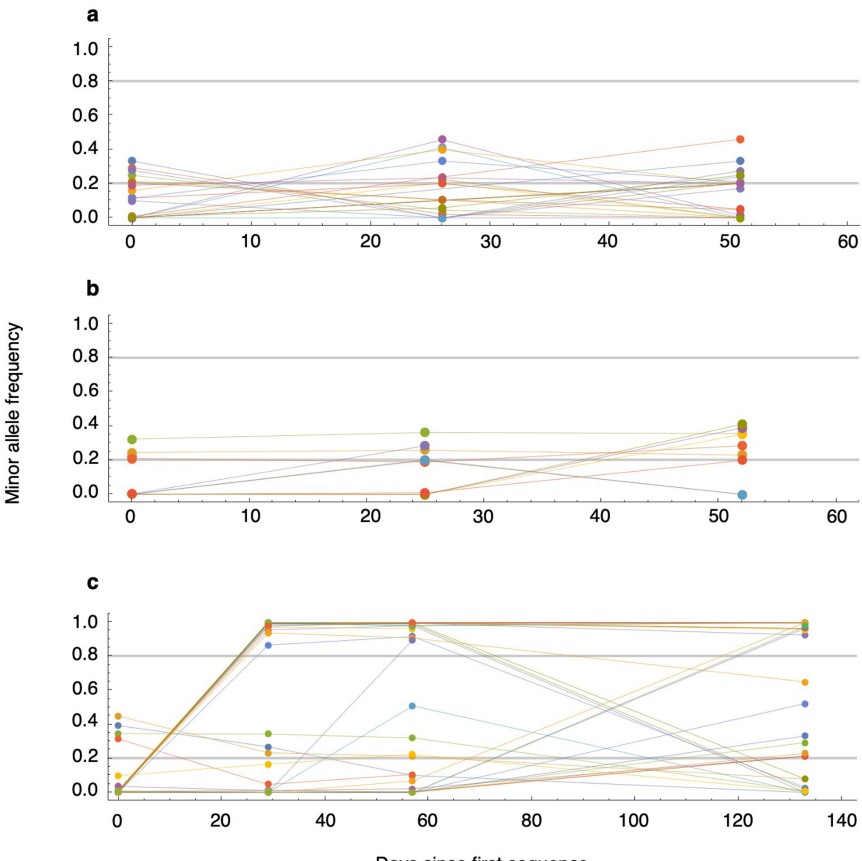

**Extended Data Fig. 7 | Dynamics of intra-host Single Nucleotide Variants (iSNVs) over time.** Temporal frequencies of iSNVs over time for (**a**,**b**) two persistent infections with zero consensus change and (**c**) a persistent infection with accelerated within-host evolution. iSNV trajectories in **a** and **b** show substantial sub-consensus activity whereby de novo mutations reach up to 40% frequency. In panel **c**, at the second time point (29 days since the first sequence), 30 consensus change mutations are detected. At the first time point, 4 iSNVs that are above 20% frequency are shared across at least one later time point. Each line represents a unique iSNV and the two horizontal grey lines represent the 20% and 80% frequency thresholds. The minimum frequency and number of bases to call an iSNV is 20% and 10 bases, respectively, and all iSNVs crossing the 20% threshold at least one more time point are included.

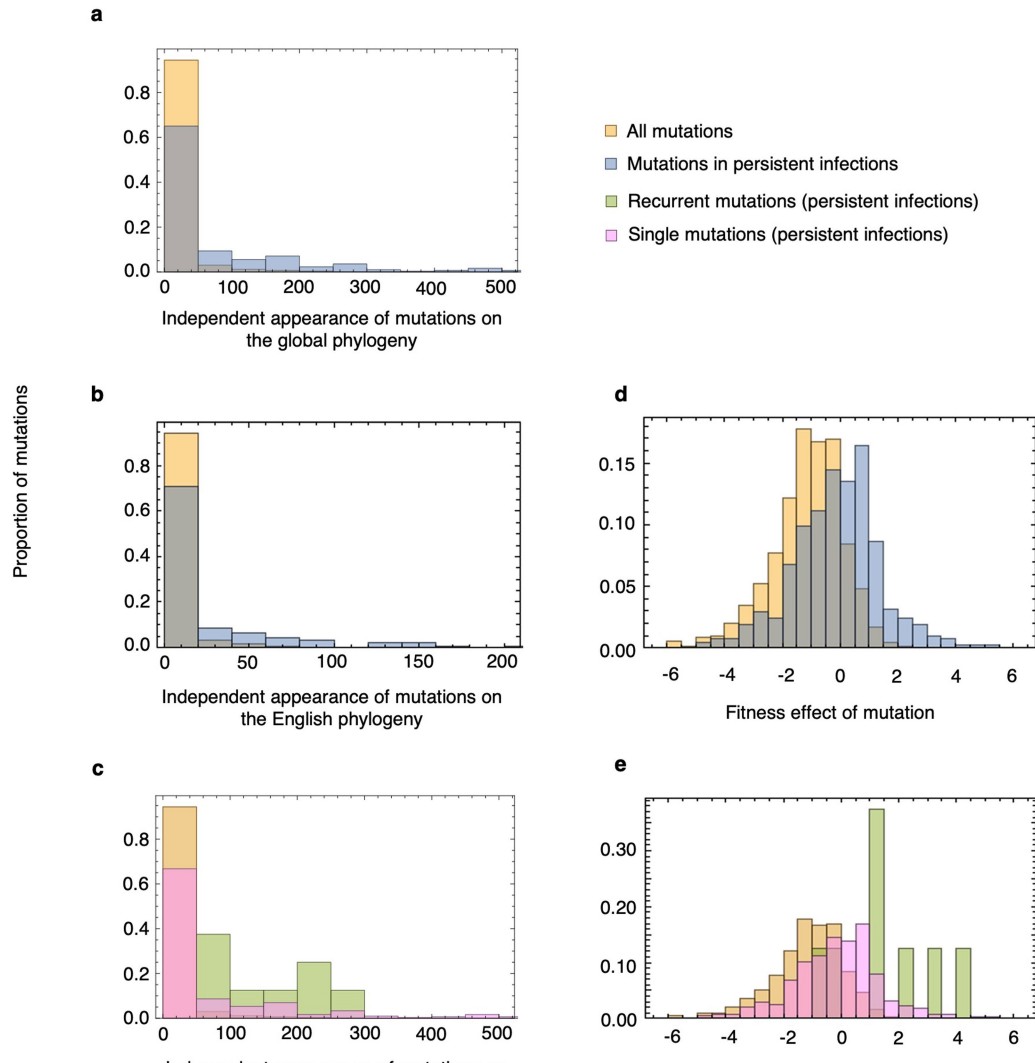

**Extended Data Fig. 8 | Counting the number of independent appearances of mutations in persistently infected individuals and their fitness effect on a global phylogeny.** (**a**–**c**) Comparing the number of independent appearances of all SARS-CoV-2 mutations (orange) on global and English phylogenies of representative samples from Alpha, Delta, BA.1, and BA.2 major lineages with mutations that are found in persistently infected individuals (blue) that only emerged in one (pink) or two (green) individuals. (**d**,**e**) Distribution of fitness effects of mutations on a globally representative phylogeny of the four major lineages of Alpha, Delta, BA.1, and BA.2. Mutations from persistent infections have an overall higher fitness than other mutations on the global phylogeny. Recurrent mutations also generally have a higher fitness than those that are found in only a single individual. Independent appearances of mutations on the global and English phylogenies are taken from https://github.com/jbloomlab/SARS2-mut-fitness/blob/main/results/mutation_counts/aggregated.csv and the fitness effect of mutations are taken from https://github.com/jbloomlab/SARS2-mut-fitness/blob/main/results/aa_fitness/aamut_fitness_by_clade.csv.

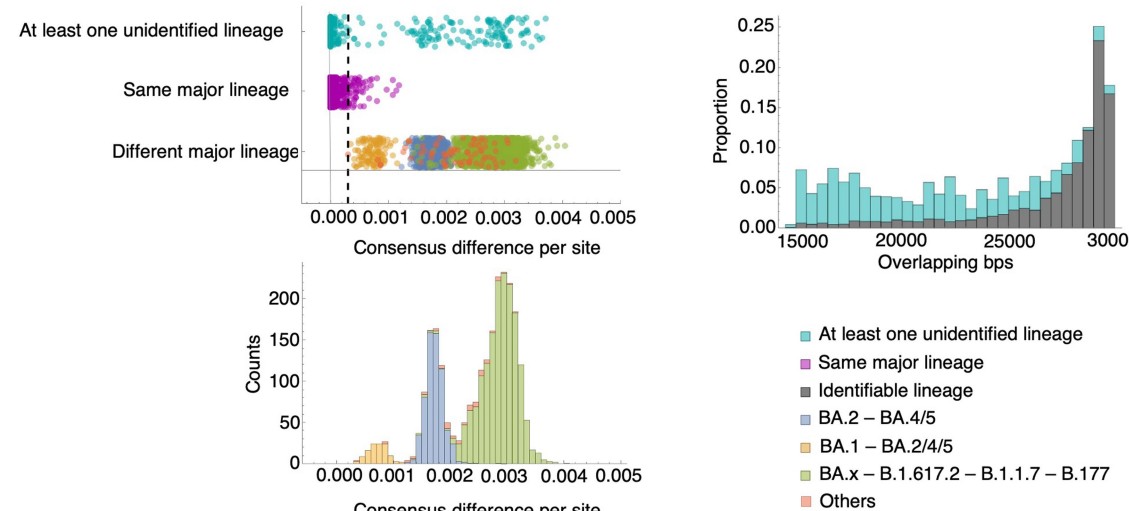

**Extended Data Fig. 9 | Pairwise differences between sequences from individuals with two or more sequences.** (Left column) Number of consensus differences per site between pairs of sequences from each individual with two or more sequences, including sequences that are less than 26 days apart. Pairs include all possible combinations of sequences from the same individual. Only sites where a nucleotide difference could be called were included. Vertical dashed line shows the lowest number of SNPs per base for pairs with different major lineages. Any pair with at least one unidentified lineage with a SNP per base smaller than the dashed line is selected as a candidate pair from a persistent infection as long as the pair is at least 26 days apart from each other. Pairs with different major lineages are coloured based on their number of SNPs per base into three groups: (i) pairs with one BA.1 and one BA.2 or BA.4 or BA.5 sequence (orange; n = 115); (ii) pairs with one BA.2 and one BA.4 or BA.5 sequence (blue; n = 628); and (iii) pairs with one Omicron (including all BA.x lineages) and one Delta (B.1.617.2), Alpha (B.1.1.7), or B.1.177 sequence (green; 1673) and (iv) all other possible combinations (red; n = 70). There was a total of 286 pairs with at least one unidentified lineage (cyan), 1470 pairs with the same major lineage (magenta), and 2486 pairs with different major lineages. (Right column) Proportion of sequences (shown in the stacked form) with different number overlapping base pairs. Those with at least one unidentified lineage (n = 286) have a lower number of overlapping base pairs relative to pairs with identifiable lineage (i.e. pairs with identical or different major lineage; n = 3956) mainly due to having lower coverage.

**Extended Data Table 1 | Baseline characteristics of SARS-CoV-2 samples from participants in this study, from 2 November 2020 to 15 August 2022**

**Baseline characteristics of all sequenced samples**

| Major lineage | Alpha | Delta | BA.1 | BA.2 |
|---|---|---|---|---|
| Infected individuals, n | 3,336 | 17,756 | 24,340 | 32,129 |
| Proportion female | 0.516 | 0.500 | 0.520 | 0.524 |
| Age*, median (min–max) | 45 (3–85) | 35 (3–85) | 45 (3–85) | 55 (3–85) |
| Sampling date (min–max) | (2020-11-2 – 2021-10-15) | (2021-4-29 – 2022-4-9) | (2022-11-30 – 2022-7-22) | (2021-12-30 – 2022-8-15) |
| SARS-CoV-2 genomes, n | 3,497 | 18,429 | 24,631 | 32,311 |
| Individuals with ≥2 sequences** | 18 | 122 | 130 | 230 |
| Persistent infections (≥26 days) | 11 | 106 | 97 | 167 |
| Reinfections (≥26 days) | 7 | 11 | 14 | 28 |

**Baseline characteristics of the persistently infected cohort**

| Major lineage | Alpha | Delta | BA.1 | BA.2 |
|---|---|---|---|---|
| Infected individuals, n | 11 | 106 | 97 | 167 |
| Proportion female | 0.4 | 0.40 | 0.41 | 0.42 |
| Age*, median (min–max) | 65 (45–85) | 45 (7–85) | 65 (7–85) | 65 (7–85) |
| Star of infection† (min–max) | (2020-12-28 – 2021-3-12) | (2021-5-31 – 2021-12-28) | (2021-12-16 – 2022-3-30) | (2022-1-27 – 2022-6-8) |

**Baseline characteristics of the non-persistently infected cohort**

| Major lineage | Alpha wave | Delta wave | BA.1 wave | BA.2 wave |
|---|---|---|---|---|
| Infected individuals, n | 5,436 | 17,139 | 25,393 | 30,934 |
| Proportion female | 0.51 | 0.50 | 0.52 | 0.53 |
| Age*, median (min–max) | 45 (2–100) | 39 (2–94) | 44 (2–99) | 56 (2–98) |
| Star of infection‡ (min–max) | (2020-12-28 – 2021-5-15) | (2021-5-16 – 2021-12-12) | (2021-12-13 – 2022-2-20) | (2022-2-21 – 2022-6-8) |

*The data is in 10-year age-bands, apart from the under 10s who are represented by two age bands.
**Total number of individuals with two or more sequences of the same lineage that are at least 26 days apart. This includes individuals with one or more undetermined lineages who have at least one sequence with determined lineage. †Start of infection is assigned on sequencing.
‡ Start of infection is assigned on calendar time since not all infections had a sequence obtained. The fixed dates broadly represent the period when each major lineage was most prevalent across the UK.

# Reporting Summary

## Statistics

For all statistical analyses, confirm that the following items are present in the figure legend, table legend, main text, or Methods section.

| n/a | Confirmed | |
|---|---|---|
| ☐ | ☒ | The exact sample size (*n*) for each experimental group/condition, given as a discrete number and unit of measurement |
| ☐ | ☒ | A statement on whether measurements were taken from distinct samples or whether the same sample was measured repeatedly |
| ☐ | ☒ | The statistical test(s) used AND whether they are one- or two-sided <br> *Only common tests should be described solely by name; describe more complex techniques in the Methods section.* |
| ☐ | ☒ | A description of all covariates tested |
| ☐ | ☒ | A description of any assumptions or corrections, such as tests of normality and adjustment for multiple comparisons |
| ☐ | ☒ | A full description of the statistical parameters including central tendency (e.g. means) or other basic estimates (e.g. regression coefficient) AND variation (e.g. standard deviation) or associated estimates of uncertainty (e.g. confidence intervals) |
| ☐ | ☒ | For null hypothesis testing, the test statistic (e.g. *F*, *t*, *r*) with confidence intervals, effect sizes, degrees of freedom and *P* value noted <br> *Give P values as exact values whenever suitable.* |
| ☒ | ☐ | For Bayesian analysis, information on the choice of priors and Markov chain Monte Carlo settings |
| ☒ | ☐ | For hierarchical and complex designs, identification of the appropriate level for tests and full reporting of outcomes |
| ☒ | ☐ | Estimates of effect sizes (e.g. Cohen's *d*, Pearson's *r*), indicating how they were calculated |

*Our web collection on statistics for biologists contains articles on many of the points above.*

## Software and code

Policy information about availability of computer code

| Data collection | No software was used for data collection. |
|---|---|
| Data analysis | For consensus sequence construction, we used shiver (v1.5.8); ARTIC Nextflow processing pipeline (v1) <br> For phylogenetic analysis, we used IQ-TREE (v1.6.12); TempEst (v1.5.3); ggtree (v3.6.2) <br> For calculation of p-values and visualisation of histogram and box plots, we used Mathematica (v13.1.0.0) |

For manuscripts utilizing custom algorithms or software that are central to the research but not yet described in published literature, software must be made available to editors and reviewers. We strongly encourage code deposition in a community repository (e.g. GitHub). See the Nature Portfolio guidelines for submitting code & software for further information.

## Data

Policy information about availability of data

All manuscripts must include a data availability statement. This statement should provide the following information, where applicable:
- Accession codes, unique identifiers, or web links for publicly available datasets
- A description of any restrictions on data availability
- For clinical datasets or third party data, please ensure that the statement adheres to our policy

All raw consensus sequences have been made publicly available as part of the COVID-19 Genomics UK (COG-UK) Consortium (https://www.cogconsortium.uk/priority-areas/data-linkage-analysis/public-data-analysis/) and are available from the European Nucleotide Archive (ENA) at EMBL-EBI under accession number

## Human research participants

Policy information about studies involving human research participants and Sex and Gender in Research.

| | |
|---|---|
| Reporting on sex and gender | Sex was determined based on self-reporting. Gender was not considered in the study design. |
| Population characteristics | Extended Data Table 1 includes the basic characteristics of the population. Extended Data Figure 1 includes a flow diagram of Office for National Statistics Covid Infection Survey (ONS-CIS) participants in this study. |
| Recruitment | The ONS-CIS is a UK household-based surveillance study in which participant households are approached at random from address lists across the UK (see Pouwels et al Lancet 2020). The survey had rolling recruitment, but in practice most recruitments occurred between September and December 2020 (see Source Data 2 in the paper and also supplementary table 4 in Vihta et al Clinical Infectious Diseases 2022). |
| Ethics oversight | The study received ethical approval from the South Central Berkshire B Research Ethics Committee (20/SC/0195). |

Note that full information on the approval of the study protocol must also be provided in the manuscript.

## Field-specific reporting

Please select the one below that is the best fit for your research. If you are not sure, read the appropriate sections before making your selection.

☐ Life sciences ☐ Behavioural & social sciences ☒ Ecological, evolutionary & environmental sciences

For a reference copy of the document with all sections, see nature.com/documents/nr-reporting-summary-flat.pdf

## Ecological, evolutionary & environmental sciences study design

All studies must disclose on these points even when the disclosure is negative.

| | |
|---|---|
| Study description | Our study identified persistent infections and reinfections with the same major lineage of SARS-CoV-2 using genomic sequence data obtained as part of the ONS-CIS. We also reported the number of symptoms and viral load dynamics for the persistent infections throughout their infection and reinfections and estimated the rate at which persistently infected individuals self-reported as having Long Covid compared to a group of non-persistently infected individuals. |
| Research sample | The study included 93,927 high-quality sequenced samples from the ONS-CIS, representing 90,146 individuals living in 66,602 households across the UK (see Extended Data Figure 1). To identify persistent infections and reinfections with the same major lineage, we limited the dataset to individuals with two or more high-quality sequences (corresponding to RT-PCR positive samples with cycle threshold ≤30), taken at least 26 days apart, and where consensus sequences were of the same major lineages of Alpha, Delta, BA.1 or BA.2. Description of the number of samples, age and sex of persistently infected individuals per major lineage is provided in Extended Data Table 1 and Extended Data Figure 1. |
| Sampling strategy | All individuals aged two years and older from each household who provide written informed consent provide swab samples (taken by the participant or parent/carer for those under 12 years), regardless of symptoms, and complete a questionnaire at assessments, which occur weekly for the first month in the survey and then approximately monthly. Participant households are approached at random from address lists across the country to provide a representative sample of the population (Pouwels et al Lancet 2020). |
| Data collection | From 26 April 2020 to 31 July 2022, assessments were conducted by study workers visiting each household; from 14 July 2022 onwards assessments were remote, with swabs taken using kits posted to participants and returned by post or courier, and questionnaires completed online or by telephone. Positive swab samples with cycle threshold ≤30 were sent for sequencing. <br><br> From February 2021, at every assessment, participants were asked "would you describe yourself as having Long Covid, that is, you are still experiencing symptoms more than 4 weeks after you first had COVID-19, that are not explained by something else?". When evaluating the probability of reporting Long Covid in persistently and non-persistently infected individuals, we considered the first assessment at least 12 weeks and at least 26 weeks after infection. <br><br> The survey offered participants the option of only having one enrollment assessment (taken by ~1%), or only assessments for one month (taken by ~1%; see Extended Data Figure 1). |
| Timing and spatial scale | For this analysis, we included data from 2nd November 2020 to 15th August 2022, spanning a period from the earliest Alpha to latest Omicron BA.2 sequences within the ONS-CIS dataset. Sample are collected from population across the UK. Status of all participants recruited in ONS-CIS per country is provided in Source Data 2. |

| Data exclusions | Only positive RT-PCR samples with Cycle threshold ≤30 (high viral titre) have been selected for sequencing. To ensure all sequences have high coverage, we only included sequences with >50% genome coverage.<br><br>The Long Covid analysis used complete cases, i.e. excluded those who did not have a response to the Long Covid in this timeframe (see Extended Data Figure 1).<br><br>All excluded data points are shown in Extended Data Figure 1. |
|---|---|
| Reproducibility | The original fasta files including consensus sequences of persistent infections and reinfections with the same major lineage as well as the other sequences used for the phylogenetic analysis are provided in the supplementary materials. |
| Randomization | Randomization was not relevant to the study design as it was based on epidemiological observations in the general population and no experimental treatment was applied. |
| Blinding | Blinding was not relevant to the study design as it was based on epidemiological observations in the general popukation and no experimental treatment was applied. |

Did the study involve field work? ☐ Yes ☒ No

# Reporting for specific materials, systems and methods

We require information from authors about some types of materials, experimental systems and methods used in many studies. Here, indicate whether each material, system or method listed is relevant to your study. If you are not sure if a list item applies to your research, read the appropriate section before selecting a response.

## Materials & experimental systems

| n/a | Involved in the study |
|---|---|
| ☒ | ☐ Antibodies |
| ☒ | ☐ Eukaryotic cell lines |
| ☒ | ☐ Palaeontology and archaeology |
| ☒ | ☐ Animals and other organisms |
| ☒ | ☐ Clinical data |
| ☒ | ☐ Dual use research of concern |

## Methods

| n/a | Involved in the study |
|---|---|
| ☒ | ☐ ChIP-seq |
| ☒ | ☐ Flow cytometry |
| ☒ | ☐ MRI-based neuroimaging |

