## [Peer Review File · Nature]

Manuscript Title: Prevalence of persistent SARS-CoV-2 in a large community surveillance study

Reviewer Comments & Author Rebuttals

Reviewer Reports on the Initial Version:

Referees' comments:

Referee #1 (Remarks to the Author):

Ghafari and colleagues use sequencing data from large scale infection survey in the United Kingdom to identify putative persistent infections and re-infections. Persistent infections have been described since the very beginning of the pandemic, both in immuno-compromised and immune competent individuals. There is a considerable number of case reports and some cohort studies, all which suggest that persistent viral shedding is not common, but also not exceedingly rare. Different studies use different definitions for persistent shedding (duration, ct-cut-offs, diagnostic procedures etc) making them difficult to compare. But many studies observed that a sizeable proportion participants shed more than 3 weeks and often observe a few that shed more than two months (here, for example: <https://erj.ersjournals.com/content/58/1/2002724>, <https://academic.oup.com/jid/article/224/8/1362/6154064>).

In contrast to these clinical studies, this study uses a large scale infection survey to estimate the prevalence persistent infections. The results are mostly in line with expectations, but this is an interesting approach using a unique resource. I have a number of major issues and concerns that might affect the validity of some results.

1) The first main conclusion is that about 1% of all infections persist for at least a month. Here, I think this number needs to be discussed i) in the light of prior evidence (consistent/surprising or not), and ii) the extent to which this is an unbiased estimate (only samples with Ct<30 are considered, etc) rather than an upper bound. Interestingly, only 1 in 7 of infections persisting for 26 days also persist for two months, suggesting a clearance rate of about 1/10days.

2) I think the argument for "non-replicating" virus is flawed. The authors argue that the fact that they do not observe a change in the consensus sequences in samples taken 4 weeks apart is evidence for some sort of dormancy since SC2 accumulates 1-2 mutations per month. I don't think this conclusion can be drawn. Every genome in the sample might have 1,2,3 or even more mutations, but if all of them are rare the consensus will not change. The comparison with cross-sectional samples in Fig S5 is not a relevant comparison (the difference between these samples depends on the TMRCAs and they have undergone many bottlenecks along their transmission chains). One could imagine sequencing cultured viruses after limiting dilution, but consensus sequence generation from a swap is unlikely to shed light on the within host evolution over a scale of a few weeks. One might see a signal in an increase of rare iSNV variation in these samples, but that requires a more careful investigation of in-sample heterogeneity and a good model of the sequencing/amplification errors, which will be challenging given that later samples tend to have high ct values.

3) If I understand correctly, 267/381 did not have a nucleotide change and the remaining 114 persistent infections had 317 nucleotide changes, thus roughly 3 per infection. The high rate of non-synonymous mutations in ORF8 probably has a lot to do with a lack of selection against non-synonymous mutations (e.g. the early stop in Alpha). The mutations that are observed as consensus changes from one time point to another are often changes that are frequently observed in cross-sectional samples as well. It would be important to characterize how dramatically these changes are over-represented compared to the expectation from cross-sectional data, for example by comparing them against relative count and fitness estimates here <https://www.biorxiv.org/content/10.1101/2023.01.30.526314v1>.

4) I assume the BA.1 infection with 33 unique mutations is p128? It looks to me as if these additional mutations are already present after 1 month. This case is indeed a bit curious, but a more thorough investigation of this case would be required to rule out that it is not due to some contamination or similar. The current brief discussion in the manuscript with a reference to Fig S2 is not helpful.

5) The reinfection analysis seems very sensitive to sample mix-up. Is there a way you could estimate the likelihood that some of the putative re-infections are due to sample swaps? Again, discussion of reinfections would benefit from a better discussion of the existing evidence (e.g. <https://www.science.org/doi/10.1126/science.abn4947> or various health-care worker cohorts). The reinfections with the same lineage are rare has been well documented -- the power of this study is the ability to look at reinfections very close in time (provided sample mix up can be ruled out) which are expected to be even rarer.

I am not going to comment on the statistical analysis of long-covid, as this is not my field of expertise.

minor:

- the title is misleading: Do you find a "high number" of infections compared to some expectation, or because the survey is very large?
- in line 267, you refer to Fig 2d but mean to refer to 3d
- the dates in the supplementary tables are in inconsistent formats. [eg. '04/03/2021' and '6/15/21']. Please use ISO dates (as you do in the sequence names).

Richard Neher

Referee #2 (Remarks to the Author):

In this manuscript, Ghafari et al. use specimens and data from community surveillance to understand the dynamics and evolution of SARS-CoV-2 in persistent infections. This is an important topic as the risk factors and consequences of these infections at the individual level are unclear. As it

has been suggested that these infections could lead to the emergence of VOC, this is an important area for further study. As stated in the abstract, the authors seek to define the viral load kinetics (as measured by RT-qPCR, the prevalence or persistent infection, and the evolution of SARS-CoV-2 in these situations).

The main contribution of the manuscript is to estimate the incidence of “persistent infection,” in this case defined as those who were positive with Ct<30 on serial samples spanning >56 days. This is an advantage of the large surveillance program from which the samples were obtained. Notably, only one infection was significantly prolonged (133 days) in the range of a number cases of very prolonged infection in immunocompromised hosts (perhaps >40 to date in published literature). Another new finding is the relationship between persistent infection and the odds of developing long COVID. I find it hard to draw much from this given the ecological nature of the study and the reliance on self reported long covid. The other analyses are interesting, but not particularly revealing about the risk factors for persistent infection or the evolution of the virus in such cases (relative to other published works).

Major points to be addressed:

1. Given the set up of the surveillance study, there is presumably data available on the individuals who had persistent infection. This would be valuable in terms of understanding whether their persistent infection is related to immunocompromising conditions or other medical comorbidity. This feels like a missed opportunity.
2. The metric for distinguishing persistent infection from reinfection is interesting. However, it isn't clear why it is needed given the phylogenetic analyses presented in the manuscript. With respect to their metric, it is interesting that there is a higher false positive rate for alpha and delta (3%) than for BA.1 and BA.2, yet the trees suggest more for BA.1 and BA.2. Presumably this is due to viral load as stated, but perhaps some commentary is needed?
3. Divergence is based on the number of consensus changes (SNPs) in serial samples. This is reasonable and falls within a “within host evolution” framework. However, in a population level study like this, it might be more appropriate to consider divergence as the distance from the most common ancestor - or branch length?
4. In ~70% of persistent infections, there were 2 successive samples (spanning a month) with no mutations accumulated. The authors speculate about decelerated evolution, given that SARS-CoV-2 accumulates 2 mutations per month in typical transmission chains. However, without evolution, it is questionable whether there is even viral replication (and therefore ongoing infection). It might reflect persistence of the RNA. The authors need to clarify this issue further as it affects their estimates of incidence and the interpretation of this number.
5. Section on emergence of notable mutations. It would be helpful to know distribution of consensus changes per infection rather than summing 277 mutations across 381 individuals (70% of which had no changes).

6. With respect to the Long COVID analysis. It seems to rely on self report of “long covid” without additional details. It is tricky to answer this question in what is essentially an ecological study. There are likely significant hidden confounders. It also might need to be more stringent in which individuals included given assessment at 12 weeks? Perhaps just the ones at 56 days? How does this fit with the reduction in symptoms in Figure 3E?

Minor points

Supplemental Figure 2 - Would be helpful if the dashed lines of the “false positive” persistent infections could be highlighted (bolded?).

Should use “RNA viral load”

Figure 1b - somehow the boxplot had shading make it hard to see the real differences. In my opinion, removing the boxplot and just showing the dots would be more helpful.

Table 1, add footnotes to indicate what percentages for reinfection indicate (i.e. numerator and denominator).

150-161 Transmission from these persistently infected individuals. Given that this is discussed, it is important to include the timing of transmission (or of the samples) since it isn't clear if this is reflective of “typical household transmission” as would happen within the first week or two of an infection vs. establishment of a persistent infection followed by late transmission.

Line 194. The use of SNP and SNV is confusing. This seems to refer to the evolutionary rate, or consensus changes (substitutions) per month. Suggest defining terms (SNP, SNV) and using consistently throughout. (SNV is sometimes interpreted to be within host variant not necessarily meeting consensus).

Figure 2a - are the mutations shared by multiple individuals more than one would expect by chance? A permutation test might be useful to assess the significance.

Lines 231-232 - Suggest consistent with units. ORF8 is 7 NS per site and spike is 61 NS total.

Referee #3 (Remarks to the Author):

This manuscript by Ghafari et al. addresses important questions about persistent SARS-CoV-2 infections as well as the association of persistent infection with Long Covid. However, the findings in the manuscript rely heavily on whether the study population is indeed representative of the general population and the completeness of the data on the study population. Currently, little to no information addressing those two topics is included in the manuscript or supplement. My specific comments are as follows:

1) This appears to be a cohort study, as a group of people were followed longitudinally for SARS-CoV-2 infection. However, the study design is not named in the manuscript. Further, there is a mix of

language used in both cohort and case-control studies. For example, table 2 has “Exposed” (cohort language) and “controls” (case-control language). The authors should appropriately state their study design and use associated language.

2) The calculation of the prevalence of persistent infections relies on several assumptions, including that the study population is representative of the general population. However, very little information has been presented on the population. The authors should present demographic information about the population in the study. The study population should be compared to the UK population to show that they are indeed similar. Further, a table comparing characteristics of households that agreed to be surveilled to those that did not agree to be surveilled would assist in assessing bias.

3) Further, data on completeness is needed to further assess bias in the study. Data on withdrawals, missing surveys, and missing samples are all needed to assess bias. In particular, data needs to be presented on the completeness of follow-up in all individuals that tested positive.

4) Given this appears to be a cohort study, an OR can be used, but an RR is a more appropriate measure. If an OR is being used because the outcome of persistent infection was assessed at the same time as the questionnaire for Long Covid and thus there is some concern around temporality, that should be clarified in the manuscript.

5) Authors should consult and follow the appropriate STROBE checklist (Checklists - STROBE (strobe-statement.org)).

Referee #4 (Remarks to the Author):

Review of 2023-01-01553
High number of SARS-CoV-2 persistent
infections uncovered through genetic analysis

March 5, 2023

This is an interesting paper that uses a unique data source to highlight the potential importance of persistent SARS-CoV-2 infection in the evolution of new variants and in Long COVID. The overall analysis appears sound and the conclusions seem reasonable, but the presentation of the methods and results requires several important clarifications that I have outlined below.

Comments

1. (line 93) The word “used” would mean exactly the same thing as “leveraged” here.
2. (lines 106-110 and Supplementary Figure 1) How many individuals had two or more RT-PCR positive samples with $Ct \leq 30$ taken at least 26 days apart and of the same lineage? It would also help to say how many such individuals there were for each major lineage considered. This would provide denominators for the numbers of persistent infections identified in Supplementary Figure 1.
3. (lines 116-117) The possible effect of the false positive ratio on the identification of persistent infections is difficult to interpret without the denominators requested in comment #2. The false positive ratio is $1 - \text{specificity}$, which is a conditional probability given that we are classifying an infection that is not actually a persistent infection. The probability that a sequence identified as a persistent infection is not actually a persistent infection could be much higher than this—it is a different conditional probability given the result of the classification. For example: When a disease is rare, most positive tests will be false positives even if the specificity of the test is high.
4. (lines 178-181) Lineage B.1.1.7 seems to be an exception to the pattern asserted here with 39% and 50% classified as reinfection for > 26 days and > 56 days, respectively.

5. (lines 274-280 and Figure 3e) The x-axis for the histogram Figure 3e seems to be backward. When the change in the number of symptoms between the first and last sequences is positive, it implies that they reported more symptoms—not fewer—at the last sequence. This contradicts the text. It is also worth pointing out that the proportion reporting more symptoms at the last sampling is much greater among the reinfections than the persistent infections (e.g., see the bump around -3 to -5 on the histogram for the reinfections).
6. (lines 295-300, lines 537-550, and Table 2) The type of regression model used to calculate adjusted odds ratios needs to be specified. The large sample size makes the Yates correction largely irrelevant, but it is not generally recommended as it tends to produce p-values and confidence intervals that are overly conservative (e.g., see Agresti 2002 or Haviland, *Statistics in Medicine* 1990). The uncorrected chi-squared test or an unadjusted regression model would produce better p-values and confidence limits.
7. (line 355) The phrase “only 9%...” would make more sense within parentheses.
8. (lines 459-460) Normally, a higher threshold implies a stricter standard for classification (e.g., in most diagnostic tests). It might help to say “prevalence threshold for defining a rare SNP” to remind readers that the higher threshold actually corresponds to a more relaxed classification (i.e., one with higher sensitivity).
9. (lines 456-465) It appears that the persistent infections identified are coming from the individuals mentioned in comment 2, and the false positive ratio is calculated based on the number of the 1,000 randomly paired lineages that are falsely identified as persistent infections. This should be stated clearly.
10. (line 467) The proportion of SARS-CoV-2 infections that persist for > 60 days is not really a prevalence. The phrase used on line 250 in the text is more accurate.
11. (lines 469-485) It would help to explain that this calculation is about the probability of obtaining a second sequenceable sample from an individual who already has produced a sequenceable sample.
12. (lines 714-716, Table 1) The table would be clearer if the columns were reordered to group i 26 days and i 56 days together. In particular, this would clarify the calculation of the percentages. It would also help to include column totals.
13. (lines 727-743, Figure 1b) These appear to be horizontal boxplots, but this is not mentioned anywhere in the text or caption. With boxplots, it

is important to specify the meaning of the whiskers. Here, they appear to extend to the minimum and maximum values.

Author Rebuttals to Initial Comments:

Response to referees

We thank all the referees for their careful read of our manuscript and their helpful and constructive comments. We have addressed all their concerns and as a result we believe that the manuscript has considerably improved.

Please find below our response to the referees' comments.

Referees' comments:

Referee #1 (Remarks to the Author):

Ghafari and colleagues use sequencing data from large scale infection survey in the United Kingdom to identify putative persistent infections and re-infections. Persistent infections have been described since the very beginning of the pandemic, both in immuno-compromised and immune competent individuals. There is a considerable number of case reports and some cohort studies, all which suggest that persistent viral shedding is not common, but also not exceedingly rare. Different studies use different definitions for persistent shedding (duration, ct-cutoffs, diagnostic procedures etc) making them difficult to compare. But many studies observed that a sizeable proportion participants shed more than 3 weeks and often observe a few that shed more than two months (here, for example: <https://erj.ersjournals.com/content/58/1/2002724>, <https://academic.oup.com/jid/article/224/8/1362/6154064>).

In contrast to these clinical studies, this study uses a large scale infection survey to estimate the prevalence persistent infections. The results are mostly in line with expectations, but this is an interesting approach using a unique resource. I have a number of major issues and concerns that might affect the validity of some results.

We thank the referee for their positive assessment of our work and for bringing these two papers to our attention. We have used them to address the comments below on percentage of prolonged persistent infections.

1) The first main conclusion is that about 1% of all infections persist for at least a month. Here, I think this number needs to be discussed i) in the light of prior evidence (consistent/surprising or not), and ii) the extent to which this is an unbiased estimate (only samples with Ct<30 are considered, etc) rather than an upper bound. Interestingly, only 1 in 7 of infections persisting for 26 days also persist for two months, suggesting a clearance rate of about 1/10days.

i) We have now compared our findings on the percentage of prolonged persistent infections with past studies which report duration of viral shedding and include a sizable number of participants. Our estimate for the proportion of infections lasting ≥ 30 days is 0.7-3.5% (now added to the manuscript). In one of the studies highlighted by the referee (<https://erj.ersjournals.com/content/58/1/2002724>), 17 (out of 92; 18%) patients who were admitted to the hospital exhibited prolonged shedding for >30 days (not clear how many of those had Ct >30 at the time of last sampling; 13 of these 17 patients were mechanically ventilated). This higher rate compared to our study is perhaps unsurprising since we do not consider hospitalised individuals, and indeed most of those in our study have few or no symptoms. In the second study, highlighted by the referee (<https://academic.oup.com/jid/article/224/8/1362/6154064>), none of the individuals had Ct ≤ 30 beyond 30 days after the onset of symptoms, and so in line with our results. We briefly discuss these papers in the discussion [line numbers: 393-399], and also emphasise throughout the manuscript that our definition of persistence *a priori* requires two or more high VL (Ct ≤ 30 samples) since we require sequence data to confirm a single infection.

ii) We are hesitant to state a clearance rate given the rebounding viral load trajectories, and particularly, we cannot be sure if or when individuals cleared their infection. Two recent papers also report observing viral trajectories with rebound both during acute (<https://elifesciences.org/articles/81849>) and prolonged infections (<https://www.nature.com/articles/s41392-022-00931-1>), and we now cite these [line numbers: 419-424].

2) I think the argument for "non-replicating" virus is flawed. The authors argue that the fact that they do not observe a change in the consensus sequences in samples taken 4 weeks apart is evidence for some sort of dormancy since SC2 accumulates 1-2 mutations per month. I don't think this conclusion can be drawn. Every genome in the sample might have 1,2,3 or even more mutations, but if all of them are rare the consensus will not change. The comparison with cross-sectional samples in Fig S5 is not a relevant comparison (the difference between these samples depends on the TMRCA and they have undergone many bottlenecks along their transmission chains). One could imagine sequencing cultured viruses after limiting dilution, but consensus sequence generation from a swap is unlikely to shed light on the within host evolution over a scale of a few weeks. One might see a signal in an increase of rare iSNV variation in these samples, but that requires a more careful investigation of in-sample heterogeneity and a good model of the sequencing/amplification errors, which will be challenging given that later sample tend to have high ct values.

We agree with the referee, although we still maintain that no consensus changes in multiple individuals after 100 days is striking since many would equate chronic/persistent infection with rapid within-host evolution, and we show that in most cases (in the individuals in our study at least) this is not the case. Nonetheless, we have removed our hypothesis that this could be due to dormancy, as we agree this is speculative. We still highlight that we see no consensus changes between many sampling time points but simply say this is likely due to little or weak selection, which we believe is an uncontroversial statement. We also cite other studies that have observed zero synonymous consensus differences between sequences of HIV for prolonged periods (<https://elifesciences.org/articles/11282>; <https://doi.org/10.1371/journal.ppat.1007167>). This required some rewriting of the section headed “Prolonged periods of stasis at the consensus level”, which was headed “evidence of non-replicating virus during infection” in the previous version [line numbers: 207-214; 407-411].

3) If I understand correctly, 267/381 did not have a nucleotide change and the remaining 114 persistent infections had 317 nucleotide changes, thus roughly 3 per infection. The high rate of non-synonymous mutations in ORF8 probably has a lot to do with a lack of selection against non-synonymous mutations (e.g. the early stop in Alpha). The mutations that are observed as consensus changes from one time point to another are often changes that are frequently observed in cross-sectional samples as well. It would be important to characterize how dramatically these changes are over-represented compared to the expectation from cross-sectional data, for example by comparing them against relative count and fitness estimates here <https://www.biorxiv.org/content/10.1101/2023.01.30.526314v1>.

We have now added a sentence in the main text explaining that the high rate of non-synonymous mutation in ORF8 could be a result of unconstrained evolution [line numbers: 252-255]. Thank you for highlighting this.

To evaluate the degree to which consensus change mutations occurring during persistent infections are representative of those found in cross-sectional data (i.e. whether these mutations are also more frequently observed in the population level), we followed the referee’s suggestion to use the data provided in this (<https://www.biorxiv.org/content/10.1101/2023.01.30.526314v1>) study. We found that these mutations are indeed found frequently at the population level, and have higher population level fitness. Those mutations appearing within two or more persistent infections were even more frequent at the population level. Of note, we observed two mutations that each occurred in two persistent infections that had low population level fitness, and we speculate this is due to differing selection pressures at the within- and

between-host levels. We have now added these results to a new section headed “Mutations observed in persistent infections occur frequently at the population-level” [line numbers: 257-282; see also supplementary figure 7] and explained the method used to extract the data [line numbers: 599-619].

4) I assume the BA.1 infection with 33 unique mutations is p128? It looks to me as if these additional mutations are already present after 1 month. This case is indeed a bit curious, but a more thorough investigation of this case would be required to rule out that it is not due to some contamination or similar. The current brief discussion in the manuscript with a reference to Fig S2 is not helpful.

The referee is correct that a significant portion of the mutations have been acquired during the first month of infection for this individual (p128). We have now carefully examined possible scenarios of contamination and coinfection for this individual.

By examining minor allele frequencies over time (below), we can rule out contamination as a likely explanation for the excess mutations during the first month since several intra-host single nucleotide variants (iSNVs) are shared across multiple timepoints. Each line on the graph corresponds to a single iSNV. We also looked for possible coinfecting variants (i.e., those that could explain the consensus changes between the first two sampling time points) by examining all of the ONS-CIS sequences but could not identify any candidates. The most likely scenario is therefore rapid within-host evolution in this individual. We now briefly describe this analysis in the main text [line numbers: 224-227; see also supplementary figure 6], with the caveat that we cannot completely rule out coinfection [line numbers: 177-180].

5) The reinfection analysis seems very sensitive to sample mix-up. Is there a way you could estimate the likelihood that some of the putative re-infections are due to sample swaps? Again, discussion of reinfections would benefit from a better discussion of the

existing evidence (e.g. <https://www.science.org/doi/10.1126/science.abn4947> or various health-care worker cohorts). The reinfections with the same lineage are rare has been well documented -- the power of this study is the ability to look at reinfections very close in time (provided sample mix up can be ruled out) which are expected to be even rarer.

The referee is right to raise concerns about the possibility of mix-ups influencing our results on identifying reinfection pairs. This is because sequences attributed to the wrong individuals, but previously infected individuals with the same major lineage, would be identified as reinfections with the same major lineage in our study. In the ONS-CIS, samples have a 3-letter code (SWT, ONS, ONC, ONW, and ONN) followed by 8 digits. The sequences similarly have a COG-ID, and there is a mapping between the two.

While mix-ups between testing and sequencing are possible, we only match exactly on barcodes and the vast majority of barcodes are negative, and even fewer still a Ct<30. Therefore, random swapping is unlikely to result in another positive with Ct<30 sent for sequencing. Positivity is 1-8% (absolute maximum, average around 3%). So, there is less than 1/30 chance of random sequenced barcode matching another positive and even lower chance for a random positive with Ct<30 to match. For each weekly sampling batch, we also checked concordance between lineage (from sequencing) and S-gene target failure (from testing), concordance between Ct (testing) and coverage (sequencing), and for the veSeq samples Ct (testing) and number of mapped reads (sequencing). We therefore have high confidence that sample mix-ups did not occur between testing and sequencing labs.

We have now explained these points in the Methods under "Identifying reinfections with the same major lineage" [line numbers: 572-580]. We also explicitly mention in the main text that sample mix-ups could elevate the reinfection rate and emphasise that the reinfections that we give are at the higher end [line numbers: 177-180].

We also agree with the referee that a unique aspect of our study is being able to identify reinfections occurring over shorter (<90 day) timescales, whereas most reinfection studies assume 90 days between positive samples to infer reinfections, including the study mentioned by the referee. Our findings are in broad agreement with recent systematic reviews showing lower rates of reinfection during the first 12 weeks since the initial infection ([https://doi.org/10.1016/S0140-6736\(22\)02465-5](https://doi.org/10.1016/S0140-6736(22)02465-5)). We have now discussed these in the main text [line numbers: 424-432].

I am not going to comment on the statistical analysis of long-covid, as this is not my field of expertise.

minor:

- the title is misleading: Do you find a "high number" of infections compared to some expectation, or because the survey is very large?

The emphasis on the high number in the title is because we believe this is the single largest study where a substantial number of persistent infections with high viral load activity have been identified. We do not find this statement to be misleading, but if the editor believes that that is the case, we are happy to modify the title accordingly.

- in line 267, you refer to Fig 2d but mean to refer to 3d

Corrected -- thank you!

- the dates in the supplementary tables are in inconsistent formats. [eg. '04/03/2021' and '6/15/21']. Please use ISO dates (as you do in the sequence names). -- Corrected

Richard Neher

Referee #2 (Remarks to the Author):

In this manuscript, Ghafari et al. use specimens and data from community surveillance to understand the dynamics and evolution of SARS-CoV-2 in persistent infections. This is an important topic as the risk factors and consequences of these infections at the individual level are unclear. As it has been suggested that these infections could lead to the emergence of VOC, this is an important area for further study. As stated in the abstract, the authors seek to define the viral load kinetics (as measured by RT-qPCR, the prevalence or persistent infection, and the evolution of SARS-CoV-2 in these situations.

The main contribution of the manuscript is to estimate the incidence of "persistent infection," in this case defined as those who were positive with Ct<30 on serial samples spanning >56 days. This is an advantage of the large surveillance program from which the samples were obtained. Notably, only one infection was significantly prolonged (133 days) in the range of a number cases of very prolonged infection in immunocompromised hosts (perhaps >40 to date in published literature). Another new finding is the relationship between persistent infection and the odds of developing long

COVID. I find it hard to draw much from this given the ecological nature of the study and the reliance on self reported long covid. The other analyses are interesting, but not particularly revealing about the risk factors for persistent infection or the evolution of the virus in such cases (relative to other published works).

We thank the referee for these comments. We agree that this is an important topic and that we are uniquely placed to determine the frequency of persistent infections.

Major points to be addressed:

1. Given the set up of the surveillance study, there is presumably data available on the individuals who had persistent infection. This would be valuable in terms of understanding whether their persistent infection is related to immunocompromising conditions or other medical comorbidity. This feels like a missed opportunity.

We agree, and the referee is not the first to ask this. Unfortunately, we are unable to link the individuals in our study with their medical records due to data protection regulations. We can say confidently, however, that these are individuals sampled within the community and, therefore, not ill enough to be hospitalised (also see our comments to Referee #1) and are indeed mostly asymptomatic. However, this does not rule out some level of immunosuppression or other factors that may make some individuals more prone to persistent infection. We have added a sentence that this is a crucial unanswered question [line number: 386-388].

2. The metric for distinguishing persistent infection from reinfection is interesting. However, it isn't clear why it is needed given the phylogenetic analyses presented in the manuscript. With respect to their metric, it is interesting that there is a higher false positive rate for alpha and delta (3%) than for BA.1 and BA.2, yet the trees suggest more for BA.1 and BA.2. Presumably this is due to viral load as stated, but perhaps some commentary is needed?

We developed our metric as a way to identify persistent infections without needing a full phylogenetic analysis, and we see this as a strength of our new method; the phylogenetic analysis was a confirmatory step. A phylogenetic method might also be possible to develop but would require sufficiently high phylogenetic resolution to establish relatedness between sequences from the same infection and the development of a clustering algorithm to identify these and to avoid calling false positives. We also note that some of the sequences from persistent infections we identified have low coverage and these sequences would not have been identified as belonging to

persistent infections using a tree-based approach. Convergent evolution could also pose a problem for tree-based approaches.

The false positive rate in our manuscript refers to the probability that two random sequences from the same major lineage would share a rare SNP by chance. The tree, however, does not show convincing evidence of false positives - all but seven of the BA.1 and BA.2 sequences from persistent infections identified by the rare SNP method cluster on phylogeny. We do not believe these are false positives, but are better explained by low coverage. All had <90% coverage, and had high Pangolin ambiguity and conflict scores (USHER version which uses phylogeny for lineage assignment), demonstrating the uncertainty on the phylogenetic tree. Indeed, two of the individuals had three sequenced virus samples where the first and last sequences cluster together while the second one (which has a high Ct and much lower coverage) does not. The probability of this pattern being explained by reinfection is very low, and the low coverage more likely explains the lack of clustering. Moreover, when viral loads are low (high Ct) the probability of falsely identifying an iSNV (including a consensus change) increases (Lythgoe et al, Science 2021). We have now more commentary on this in the main text [line numbers: 139-143].

3. Divergence is based on the number of consensus changes (SNPs) in serial samples. This is reasonable and falls within a “within host evolution” framework. However, in a population level study like this, it might be more appropriate to consider divergence as the distance from the most common ancestor - or branch length?

We agree that counting consensus differences is an appropriate measure of divergence in this context. By “considering the distance from the most common ancestor” we think the referee here is referring to the most recent common ancestor (MRCA) of each infection. We could do this, but in practice the MRCA and the consensus of the first sample are either identical (in most cases) or have 1-2 consensus differences. Hence, this would only have a very minor effect on our results, and would have no effect on the proportion of samples with no consensus changes (i.e., a branch length of zero).

4. In ~70% of persistent infections, there were 2 successive samples (spanning a month) with no mutations accumulated. The authors speculate about decelerated evolution, given that SARS-CoV-2 accumulates 2 mutations per month in typical transmission chains. However, without evolution, it is questionable whether there is even viral replication (and therefore ongoing infection). It might reflect persistence of the RNA. The authors need to clarify this issue further as it affects their estimates of incidence and the interpretation of this number.

Interestingly, referee #2 is arguing that lack of consensus change is due to lack of viral replication, whereas referee #1 suggests it is consistent with limited or no directional selection. We have now addressed this (see response to referee 1 comment #2). We find it unlikely that the lack of consensus change is due to persistence of RNA because this cannot explain the high, and rebounding, viral loads we observe; a high VL sample with Ct <30 is required for sequencing [line numbers: 298-304].

5. Section on emergence of notable mutations. It would be helpful to know distribution of consensus changes per infection rather than summing 277 mutations across 381 individuals (70% of which had no changes).

We thank the referee for their suggestion. We have now added a new panel to Figure 3 (panel c) showing the distribution of consensus differences per site between all samples of every persistent infection.

6. With respect to the Long COVID analysis. It seems to rely on self report of “long covid” without additional details. It is tricky to answer this question in what is essentially an ecological study. There are likely significant hidden confounders. It also might need to be more stringent in which individuals included given assessment at 12 weeks? Perhaps just the ones at 56 days? How does this fit with the reduction in symptoms in Figure 3E?

As a clinical outcome, it is not possible to investigate associations with Long Covid other than through observational analyses (termed ecological analyses by the referee), since it is not possible to experimentally allocate participants to get Long Covid or not. There may always be residual confounders of any observational analysis, but we controlled for many that could potentially influence results, including those identified in previous analyses of self-reported Long Covid in the survey (<https://bjgp.org/content/71/712/e806>). All of the possible confounders we controlled for are listed in the methods, and we now also added the potential for unmeasured confounding as a limitation of this analysis [line numbers: 708-710].

The 12-week threshold is used to define Post-COVID-19 Condition in UK clinical guidelines and Post COVID-19 Condition by the World Health Organisation, hence we adopted this nationally and internationally recognised case definition in our analysis. We have also added a limitation regarding the use of self-reported data for classifying Long Covid in the methods section [line numbers: 670-674].

Minor points

Supplemental Figure 2 - Would be helpful if the dashed lines of the “false positive” persistent infections could be highlighted (bolded?).

We do not think there is good evidence these are false positives, and are more likely explained by low coverage samples (see response to comment above).

Should use “RNA viral load” – Corrected.

Figure 1b - somehow the boxplot had shading make it hard to see the real differences. In my opinion, removing the boxplot and just showing the dots would be more helpful.

We have now modified the figure.

Table 1, add footnotes to indicate what percentages for reinfection indicate (i.e. numerator and denominator). -- Added. Thank you!

150-161 Transmission from these persistently infected individuals. Given that this is discussed, it is important to include the timing of transmission (or of the samples) since it isn't clear if this is reflective of “typical household transmission” as would happen within the first week or two of an infection vs. establishment of a persistent infection followed by late transmission.

In the interests of saving space, we have now removed this paragraph from the manuscript.

Line 194. The use of SNP and SNV is confusing. This seems to refer to the evolutionary rate, or consensus changes (substitutions) per month. Suggest defining terms (SNP, SNV) and using consistently throughout. (SNV is sometimes interpreted to be within host variant not necessarily meeting consensus).

We have now ensured that we use a consistent terminology throughout the text. We only use the word SNP in reference to consensus differences between individual sequences and a population-level consensus for each major lineage. We refrain from using the word SNP when discussing consensus changes per month. We also replaced SNV with the number of consensus differences between randomly selected pairs of sequences from the population.

Figure 2a - are the mutations shared by multiple individuals more than one would expect by chance? A permutation test might be useful to assess the significance.

This is similar to comment #3 from referee 1; please see our response to referee 1.

Lines 231-232 - Suggest consistent with units. ORF8 is 7 NS per site and spike is 61 NS total.

We have now given the per base rate for both in the text.

Referee #3 (Remarks to the Author):

This manuscript by Ghafari et al. addresses important questions about persistent SARS-CoV-2 infections as well as the association of persistent infection with Long Covid. However, the findings in the manuscript rely heavily on whether the study population is indeed representative of the general population and the completeness of the data on the study population. Currently, little to no information addressing those two topics is included in the manuscript or supplement. My specific comments are as follows:

1) This appears to be a cohort study, as a group of people were followed longitudinally for SARS-CoV-2 infection. However, the study design is not named in the manuscript. Further, there is a mix of language used in both cohort and case-control studies. For example, table 2 has “Exposed” (cohort language) and “controls” (case-control language). The authors should appropriately state their study design and use associated language.

The underlying study from which episodes of infection in the community population are drawn, the ONS COVID-19 Infection Survey (ONS-CIS), is, as first described in Pouwels et al [1], a longitudinal cohort study, sending invitations to participate to households that are either randomly selected from address lists -- this is the method for recruiting the vast majority of participants; response rates 12-14% (links in Table 2 of <https://www.ons.gov.uk/peoplepopulationandcommunity/healthandsocialcare/conditionsanddiseases/datasets/covid19infectionsurveytechnicaldata>), or in Northern Ireland and between April-June 2020, only in England and Wales those participating in previous surveys who had agreed to take part (response rates ~50%, same reference). This underlying cohort is broadly representative of the UK population (for example, see Supplementary Tables 3-6 in Vihta et al [2]). However, the current analysis identifies and compares persistent infections and reinfections identified from positive swab tests taken within the cohort study. Similarly, for the Long Covid analysis, we compare the prevalence of Long Covid in the persistently infected individuals relative to those who are not persistently infected where both groups are identified through positive swab tests taken within the study. We have now provided further detail on the ONS-CIS in the

methods section [line numbers: 470-472; 482-484] and clarified the language throughout.

References

1. Pouwels, K. B. et al. Community prevalence of SARS-CoV-2 in England from April to November, 2020: results from the ONS Coronavirus Infection Survey. *Lancet Public Health* 6, e30–e38 (2021).

2. Vihta KD, Pouwels KB, Peto TEA, et al. Symptoms and Severe Acute Respiratory Syndrome Coronavirus 2 (SARS-CoV-2) Positivity in the General Population in the United Kingdom. *Clin Infect Dis.* 2022;75(1):e329-e337

2) The calculation of the prevalence of persistent infections relies on several assumptions, including that the study population is representative of the general population. However, very little information has been presented on the population. The authors should present demographic information about the population in the study. The study population should be compared to the UK population to show that they are indeed similar. Further, a table comparing characteristics of households that agreed to be surveilled to those that did not agree to be surveilled would assist in assessing bias.

Rather than repeating this information, which is already published, we now refer to characteristics of the COVID-19 Infection Survey population and the UK population in Supplementary Tables 3-6 in Vihta et al [2]. As the vast majority of recruitment comes from invitations sent to households randomly selected from address lists in whom we do not have relevant demographic information, we are not able to compare characteristics of those agreeing and not agreeing to participate. We have added a statement in the methods about this limitation of the study [line numbers: 473-476].

3) Further, data on completeness is needed to further assess bias in the study. Data on withdrawals, missing surveys, and missing samples are all needed to assess bias. In particular, data needs to be presented on the completeness of follow-up in all individuals that tested positive.

We have added a precis of the following details to the Methods [line numbers: 486-500] and provided more detailed information on recruited participants and completeness of the follow-ups in Supplementary Tables 3 and 4.

To date, of 535,731 participants recruited into the COVID-19 Infection Survey, 109,417 (20%) have either completed their participation or withdrawn. It is important to note that

the survey design offered participants the option of only having one enrolment assessment (taken by ~1%), or only assessments for one month (taken by ~1%), that moving house is a major reason for completing participation in the survey (since this leads to participants no longer being eligible for follow-up since it is the original address that is sampled), that a small number of participants have died (0.4%) ending their participation, and that in July 2022 the survey moved to a remote data collection approach at which point some participants chose to end their participation. Given these factors, and the fact that most participants were recruited September-December 2020, we consider 20% withdrawal/completion is reasonably high follow-up.

In terms of missing surveys, the cohort design is to sample individuals on a 28-42 day cycle, in order to achieve a total target number of swab samples per month (as per the approved protocol on <https://www.ndm.ox.ac.uk/covid-19/covid-19-infection-survey/protocol-and-information-sheets>), and also reflecting the availability of study workers to conduct home visits all over the UK. Participants could also choose to reschedule visits to a more convenient time if necessary. The research dataset contains only details of assessments that were conducted, not that were scheduled but missed, and therefore, given the variable time between visits, it is not straightforward to provide a simple metric of missed assessments. However, for the time period considered to identify persistent infections and reinfections in this analysis, 2 November 2020 to 15 August 2022, 96.20% of swabs had a negative result and 1.88% a positive result (1.92% were void). For those with positive test results, the mean time since the previous assessment was 35.2 days and to the next assessment was 37.1 days. For those with a negative test, the associated numbers are 31.8 and 33.0 days. There was no statistical difference in the time between sampling for individuals with persistent infection compared to those testing positive. See Supplementary Table 4 for a full description of distributions.

4) Given this appears to be a cohort study, an OR can be used, but an RR is a more appropriate measure. If an OR is being used because the outcome of persistent infection was assessed at the same time as the questionnaire for Long Covid and thus there is some concern around temporality, that should be clarified in the manuscript.

As described above, whilst the underlying study design is a cohort study, this specific analysis of Long Covid focuses on comparing persistent vs non-persistent infections in terms of the risk of subsequent self-reported Long Covid (binary outcomes, at least 12 weeks and at least 26 weeks following the first positive test). As these are binary outcomes rather than a time-to-event outcome, either an OR or a RR could be used, and there is no strong reason to prefer one over the other [line numbers: 683-689].

5) Authors should consult and follow the appropriate STROBE checklist (Checklists - STROBE (strobe-statement.org)).

As described above, the underlying study design is a cohort study, but this specific analysis focuses on comparing persistent vs reinfections identified from positive swab samples taken from the underlying population. It is therefore neither a case-control study nor a cohort study in typical terminology. We would be happy to follow editorial guidance on appropriate checklists.

Referee #4 (Remarks to the Author):

This is an interesting paper that uses a unique data source to highlight the potential importance of persistent SARS-CoV-2 infection in the evolution of new variants and in Long COVID. The overall analysis appears sound and the conclusions seem reasonable, but the presentation of the methods and results requires several important clarifications that I have outlined below.

Thank you for these comments.

Comments

1. (line 93) The word “used” would mean exactly the same thing as “leveraged” here.

We have now modified this line.

2. (lines 106-110 and Supplementary Figure 1) How many individuals had two or more RT-PCR positive samples with $Ct \leq 30$ taken at least 26 days apart and of the same lineage? It would also help to say how many such individuals there were for each major lineage considered. This would provide denominators for the numbers of persistent infections identified in Supplementary Figure 1.

This is an important point, and we thank the reviewer for raising it. The number of candidate persistent infections is not that large: 18 Alpha, 122 Delta, 130 BA.1, and 230 BA.2. We now present these numbers in the main text [line numbers: 115-118] and Supplementary Table 1. In addition, we revised Supplemental Figure 1 and its associated legend so that these numbers can be visualised in context. We would also like to clarify that Supplementary Figure 1 was originally intended to show all candidate persistent infections of *any duration* (including those that are shorter than 26 days). However, after reflecting on the referee’s comments here and the one below, we believe

a more appropriate plot would be one that shows the number of persistent infections with $Ct \leq 30$ taken at least 26 days apart and of the same lineage (including those with undetermined lineages), and hence the figure directly shows the number of candidate persistent infections as it pertains to our definition in the main text.

3. (lines 116-117) The possible effect of the false positive ratio on the identification of persistent infections is difficult to interpret without the denominators requested in comment #2. The false positive ratio is $1 - \text{specificity}$, which is a conditional probability given that we are classifying an infection that is not actually a persistent infection. The probability that a sequence identified as a persistent infection is not actually a persistent infection could be much higher than this—it is a different conditional probability given the result of the classification. For example: When a disease is rare, most positive tests will be false positives even if the specificity of the test is high.

The referee highlights an important point. We now report the number of candidate persistent infections (see Supplementary Table 1). This is low - a total of 500 candidates - so with a false positive rate of 1% only 5 out of the identified 381 persistent infections (1.3%) would be false positives). We therefore do not have a conditional probability problem here, but the referee was right to point this out. We believe that reporting the number of candidate infections, as suggested by the reviewer, addresses this point.

4. (lines 178-181) Lineage B.1.1.7 seems to be an exception to the pattern asserted here with 39% and 50% classified as reinfection for > 26 days and > 56 days, respectively.

One of the reasons why we found fewer persistent infections and reinfections with Alpha is because fewer positive samples with $Ct \leq 30$ were sent for sequencing and successfully sequenced, particularly during the early stages of the emergence of Alpha (see Supplementary Figure S6 in Lythgoe et al. 2022 <https://doi.org/10.1101/2022.01.05.21268323>). The low numbers for both may affect the percentages and bias the results, as could other factors such as the timing of vaccination rollout. We now briefly mention this in the manuscript [line numbers: 197-201].

5. (lines 274-280 and Figure 3e) The x-axis for the histogram Figure 3e seems to be backward. When the change in the number of symptoms between the first and last sequences is positive, it implies that they reported more symptoms—not fewer—at the last sequence. This contradicts the text. It is also worth pointing out that the proportion reporting more symptoms at the last sampling is much greater among the reinfections

than the persistent infections (e.g., see the bump around -3 to -5 on the histogram for the reinfections).

We agree that the word “change” in panels d and e reflect two different frames of reference, and nonetheless is open to misinterpretation. To avoid confusion, we have now modified the x-axis for both panels d and e to make the calculation we did more explicit and clarified what it represents in the figure caption. Thank you for highlighting that a noticeable fraction of reinfections report more symptoms at the last sampling time point compared to persistent infections. We have now highlighted this in our results [line numbers: 340-341].

6. (lines 295-300, lines 537-550, and Table 2) The type of regression model used to calculate adjusted odds ratios needs to be specified. The large sample size makes the Yates correction largely irrelevant, but it is not generally recommended as it tends to produce p-values and confidence intervals that are overly conservative (e.g., see Agresti 2002 or Haviland, Statistics in Medicine 1990). The uncorrected chi-squared test or an unadjusted regression model would produce better p-values and confidence limits.

We thank the referee for noticing this. We had inadvertently included an older version of one part of the analysis, and the reference to the Chi square test with Yates correction was incorrectly retained. We have updated the results to the correct version and assure the referee that all quoted p-values are from unadjusted or adjusted logistics regression models.

7. (line 355) The phrase “only 9%. . .” would make more sense within parentheses.

Now revised.

8. (lines 459-460) Normally, a higher threshold implies a stricter standard for classification (e.g., in most diagnostic tests). It might help to say “prevalence threshold for defining a rare SNP” to remind readers that the higher threshold actually corresponds to a more relaxed classification (i.e., one with higher sensitivity).

Thank you for raising this point. We would rather not use the word “prevalence”, since prevalence typically refers to a proportion, whereas the threshold referred to in our manuscript is the number of sequences. However, we have reworded the text to make it clear that the threshold we are referring to is number of sequences.

9. (lines 456-465) It appears that the persistent infections identified are coming from the individuals mentioned in comment 2, and the false positive ratio is calculated based on

the number of the 1,000 randomly paired lineages that are falsely identified as persistent infections. This should be stated clearly.

This is correct and we agree. We have now clarified this in response to comment 2 and Supplementary Figure 1 caption.

10. (line 467) The proportion of SARS-CoV-2 infections that persist for > 60 days is not really a prevalence. The phrase used on line 250 in the text is more accurate.

We have now modified this throughout the text to keep in line with the phrase used in line 250 of the original version, “the percentage of SARS-CoV-2 infections that are persistent and last for longer than 60 days”.

11. (lines 469-485) It would help to explain that this calculation is about the probability of obtaining a second sequenceable sample from an individual who already has produced a sequenceable sample.

We agree, and have now explained this [line numbers: 629-635].

12. (lines 714-716, Table 1) The table would be clearer if the columns were reordered to group ≤ 26 days and ≤ 56 days together. In particular, this would clarify the calculation of the percentages. It would also help to include column totals. -- Now revised.

13. (lines 727-743, Figure 1b) These appear to be horizontal boxplots, but this is not mentioned anywhere in the text or caption. With boxplots, it is important to specify the meaning of the whiskers. Here, they appear to extend to the minimum and maximum values.

Referee #2 also raised an issue with the box-whisker plot in Figure 1b. We have taken out the boxes altogether and only present individual points to enhance readability.

Reviewer Reports on the First Revision:

Referees' comments:

Referee #1 (Remarks to the Author):

The authors have addressed my main issue -- the claim that there is evidence for non-replicating virus -- by removing this claim. Instead, they now discuss this as 'stasis at the consensus level', which is fine though not unexpected.

The individual with 33 fixed mutations over 133 days of infection is now discussed at some level of detail as an example of a strong signal of adaptation. After looking at the consensus sequences from that individual, I have the impression that the large number of changes might be due to anomalous mutation processes, for example through molnupiravir treatment. At least 17 of these mutations are G->A mutations and there are only few C->T mutations. This is very unusual for SARS-CoV-2, but expected when treated with molnupiravir. See this paper for a careful analysis of mutation spectrum induced by molnupiravir:

<https://www.medrxiv.org/content/10.1101/2023.01.26.23284998v2>

I remain unconvinced that this (very interesting) case is typical and don't think it should be discussed as evidence for strong selection. Beyond this case, it is nice to see how mutations observed multiple times in persistent infections have the highest fitness effect estimates in cross-sectional analysis. This is certainly very suggestive of an within host adaptive effect, but could also be driven by anomalously high mutation rates at these sites.

Minor points:

* Abstract: I would replace 'little or weak selection' with 'limited within-host adaptation'. There is a lot of purifying selection

* the discussion of mutations and their fitness (lines 267--275) could use some more careful language. Mutations don't "have fitness", but they affect fitness of the virus. Phrasing this as 'effect on fitness' or 'fitness cost' might be better.

Richard Neher

Referee #2 (Remarks to the Author):

I appreciate the authors' efforts to address my comments and those of the other reviewers. Overall, I

think the manuscript is improved. There are still issues with the strength of the data in places (in terms of well some claims are justified) but the authors appear to have pushed the data as far as they reasonably can.

I am still troubled by the framing, especially on lines 381-391. Yes, there is lots of evidence for chronic infections of immunocompromised hosts being the source of highly divergent VOC. But the numbers that the authors base their “striking” incidence estimate on includes the 70% of individuals who had no measurable within-host evolution (and therefore would not lead to highly divergent variants). Only a couple individuals had infections lasting >100 days and significant divergence within-hosts (as far as I can tell). I think it is more accurate to say that they have identified the likely incidence of people with prolonged detection of significant amounts of viral RNA. How this relates to ongoing replication of the virus, host immune status, and the number of people who could be incubating highly divergent variants is unclear from the data presented.

As I suggest above, I still find the absence of detectable consensus changes in the vast majority of the persistent infections perplexing. It is not out of line with what reviewer 1 commented (which was focused on how within host rates were undercounted for based on consensus changes). Given the available case reports (of which there are now many) show rates of consensus changes >2-3 per month, what are we to think then of majority with no consensus changes? This is somewhat important as it affects the overall incidence estimate.

Lines 404-406 - Elevated or accelerated (or “rapid” as is used in some places) within-host rates in persistent infection are by comparison to SARS-CoV-2 evolutionary rates along a chain of transmission, in which bottlenecks slow down the process of mutation accumulation. Somehow this should be clarified, lest the reader think that a persistent and non-persistent infection somehow have different within-host evolutionary rates (which may or may not be the case).

Reviewer 1 pointed out something that I missed on the initial review - the one individual with the large number of NS changes in the first 30 days (20 I think). This is really perplexing as it is 10x the typical evolutionary rate and much higher (perhaps 4-5x) the rates of mutation accumulation in immunocompromised hosts (from case reports). The most obvious explanation would be contamination during sequencing - but this seems less likely given the trajectories shown. Another possibility would be molnupiravir treatment, which increases the mutation rate. But the authors report they have no access to these data. Finally, APOBEC activity would be another possibility. It might be possible to sort this out by looking at the mutations and contexts closely. The current explanation - there’s just rapid evolution - feels a little “hand wavy” for something that is so far from what has been reported.

Referee #3 (Remarks to the Author):

The authors partially addressed my concerns. They have at least clarified what they did enough that one can determine the study design. This is a cohort study. They have taken a cohort study, reduced the

study to look at those that were SARS-CoV-2 infected over the time period, assessed their main exposure (persistently infected or not persistently infected), and then followed them forward for their outcome (long COVID).

1) The appropriate checklist given this is a cohort checklist which can be found <https://www.strobe-statement.org/checklists/>

2) It is still not possible to assess potential bias due to incomplete surveys or samples. A flow chart which starts with all the infections that were identified between 2nd November 2020 to 15th August 2022 should be included (supplement is fine). The flow chart should specify who was not included in the analysis due to a lack of a follow-up sample or missing survey. The methods should also specify how that was handled as well. For example, what did the authors do if a participant was positive but failed to have a follow-up swab within the time necessary to determine that they were shedding persistently? For example, what if they were positive but didn't have another swab or their next swab was 45 days or more out and negative? What do the authors do if a participant was missing the follow-up survey to determine if they still had symptoms at 28 days or further out? Was missingness consistent between the two groups? The authors provided information about the overall cohort. Still, here we really need information on the subcohort that was infected and should have been included in this analysis if the world were perfect (everyone completed every sample/survey etc), including how missing data was handled and who wasn't included due to missing data.

Referee #4 (Remarks to the Author):

This study uses genetic sequences from a large surveillance study of UK households to study persistent infections and reinfections with SARS-CoV-2. The combination of the data source and the approach is unique and generates a number of useful insights into the role of persistent infections in the evolution and epidemic dynamics of the virus.

My previous comments have been addressed adequately, and the methods and results are now described clearly. I have only a few typo corrections and a two minor suggestions about the description of the statistical methods.

1. (line 362) The "lower rate of reporting Long COVID" is actually a lower probability.
2. (lines~357-366) The type of regression model used to adjust for possible confounders is never named. It would be good to specify that it was a logistic regression model either here or in lines~693-706.
3. (line 472) "supplementary table~4" needs to be capitalized, and "ref 45" should not be a superscript.
4. (lines~693-706) The description of the logistic regression model is clear, but no evaluation of goodness of fit (e.g., Hosmer-Lemeshow test or calibration belt) is described. It would be good to add this.

Author Rebuttals to First Revision:

Response to referees

We thank the editor and all the referees for their careful read of our manuscript and their helpful comments. We are very pleased that Referees 1, 3 and 4 are happy with the manuscript and with only a few helpful comments that we have now addressed. We are also pleased that we have addressed many of the comments from Referee 2. Here we have outlined our responses to the editor and referees' comments, and how we have revised the manuscript where appropriate.

Editor comments

The referees continue to raise some concerns with the framing and with the strength of the analysis. In particular, they feel that you cannot draw conclusions about "persistent infection"; only about persistent viral RNA. They feel this detracts from the significance as, if the virus is not replicating or infectious, it will likely not give rise to divergent lineages or act as a viral reservoir. As such, the significance may rest on the association with persistent symptoms, but here the referees remain concerned about potential confounders and bias. Referee 3 continues to feel that the STROBE checklist should be completed and more information provided about sources of bias and missing data.

If you are willing to reframe the data, and address the remaining concerns we would be willing to discuss your responses with the reviewers in one further round of review, however, given the nature of the concerns we cannot predict the outcome.

In our response below and in the manuscript, we have provided further evidence on why we believe virus sequences obtained from the persistently infected individuals within our dataset belong to replication-competent viruses. We also provided the STROBE checklist for cohort studies and a flow diagram as requested by Referee 3. We emphasise that although we use data collected as part of a surveillance study with rolling recruitment, most of our results involve genetic analysis of infections from individuals with multiple sequences, and therefore is not a cohort study as such. It is only the Long Covid comparison, which constitutes a small part of the paper, that could be considered a cohort analysis. Therefore, complying with all the requirements laid out in the STROBE cohort checklist, including changing the title of the manuscript to explicitly mention the word 'cohort' may not be appropriate. We would be happy to follow the Editors' guidance on the most appropriate way to deal with this, since most of the study is evolutionary in nature, and therefore does not fall within a traditional epidemiological framework.

Referee #1 (Remarks to the Author):

The authors have addressed my main issue -- the claim that there is evidence for non-replicating virus -- by removing this claim. Instead, they now discuss this as 'stasis at the consensus level', which is fine though not unexpected.

We are pleased that the referee's concerns have been addressed, and we thank the referee for encouraging us to articulate this better. In light of the comments of Referee 2, we have also now briefly discussed why stasis at the consensus level over short timescales is not unexpected, as the referee rightly points out.

We have also included two examples of iSNV trajectories for persistently infected individuals with zero consensus change over a period of at least 50 days or longer with substantial subconsensus evolutionary activity (p66 and p133). This provides further support to the statements made by the referee in the earlier round of revisions, and we thank them for bringing them to our attention.

The individual with 33 fixed mutations over 133 days of infection is now discussed at some level of detail as an example of a strong signal of adaptation. After looking at the consensus sequences from that individual, I have the impression that the large number of changes might be due to anomalous mutation processes, for example through molnupiravir treatment. At least 17 of these mutations are G->A mutations and there are only few C->T mutations. This is very unusual for SARS-CoV-2, but expected when treated with molnupiravir. See this paper for a careful analysis of mutation spectrum induced by molnupiravir:

<https://www.medrxiv.org/content/10.1101/2023.01.26.23284998v2>

I remain unconvinced that this (very interesting) case is typical and don't think it should be discussed as evidence for strong selection. Beyond this case, it is nice to see how mutations observed multiple times in persistent infections have the highest fitness effect estimates in cross-sectional analysis. This is certainly very suggestive of an within host adaptive effect, but could also be driven by anomalously high mutation rates at these sites.

We agree with the referee that this persistent infection is an extreme case and may not represent an example of strong selection. Nevertheless, as the referee also pointed out, we still think this is an interesting case. Therefore, we chose to keep it in the text but not as an example of strong adaptive evolution. Unfortunately, we do not have any information on whether these individuals have received any treatments (mentioned in the discussion as part of our limitation in this study). We have now speculated on

whether they might be under molnupiravir treatment as Referee 2 also suggested. We thank the referee for bringing that paper to our attention.

Minor points:

* Abstract: I would replace 'little or weak selection' with 'limited within-host adaptation'. There is a lot of purifying selection -- Now revised

* the discussion of mutations and their fitness (lines 267--275) could use some more careful language. Mutations don't "have fitness", but they affect fitness of the virus. Phrasing this a `effect on fitness` or `fitness cost` might be better.

Thanks for highlighting this. This is now revised.

Richard Neher

Referee #2 (Remarks to the Author):

I appreciate the authors' efforts to address my comments and those of the other reviewers. Overall, I think the manuscript is improved. There are still issues with the strength of the data in places (in terms of well some claims are justified) but the authors appear to have pushed the data as far as they reasonably can.

I am still troubled by the framing, especially on lines 381-391. Yes, there is lots of evidence for chronic infections of immunocompromised hosts being the source of highly divergent VOC. But the numbers that the authors base their "striking" incidence estimate on includes the 70% of individuals who had no measurable within-host evolution (and therefore would not lead to highly divergent variants). Only a couple individuals had infections lasting >100 days and significant divergence within-hosts (as far as I can tell).

The statement that there is no measurable within-host evolution in 70% of the individuals is not quite correct. We observed that 68% (259/381) of individuals displayed no *consensus* change during their infection. Of these, 93% (242/259) were less than 2 months long, which is not surprising in the absence of strong directional selection. On the other hand, however, there were 17 persistent infections with three or more sequences of which the first two sequences (typically about a month apart) had zero consensus differences, but, crucially, 41% (7/17) gained a consensus change later in the infection. This suggests that even among persistent infections with at least one pair of samples with zero consensus difference, the virus evolves measurably at the consensus level as time progresses since the onset of infection which is in line with the

comment from Referee 1 that it takes time for *de novo* mutations to increase in frequency during infection. Moreover, among the remaining 59% (10/17) with no consensus change, we often find significant sub-consensus activity with iSNV frequencies going up to high frequencies (~40%) and returning to below 5% at a later time point indicating that the virus population is indeed replicating during infection. Among persistent infections with two sampled time points, we observed comparable trajectories. We have now added two example iSNV trajectories for such persistent infections in the supplementary materials (see Supplementary Figure 7).

It appears that the referee is equating persistent infections with accelerated within-host evolution of the virus. However, as we explained in the discussion, one of the key take home points of our paper is that for individuals in the community who have long infections we should not assume the virus is always under strong selection, particularly if they are not under treatment to clear the virus and, therefore, no strong selection on the virus to evolve. Similarly, at the population level, within major lineages, most observed evolution is neutral or nearly neutral.

I think it is more accurate to say that they have identified the likely incidence of people with prolonged detection of significant amounts of viral RNA. How this relates to ongoing replication of the virus, host immune status, and the number of people who could be incubating highly divergent variants is unclear from the data presented.

We disagree with the referee's assessment that our data only shows the detection of persistent viral RNA in individuals but not ongoing replication for a number of reasons:

1- The 10-100+ fold increases in viral load that we observe during persistent infections are inconsistent with the virus not replicating. In response to the referee's comment we had already added this to the earlier revised version:

"Overall, 55/67 (82%) of persistent infections where we had sufficient data showed a resurgence in viral load after an initial drop (see Supplementary Figure 4a). These rebounding viral load dynamics also support the presence of replicating viruses during these infections."

The referee appears to have missed this statement when reviewing the revised manuscript. We emphasised this point both in the results and discussion.

2- There are numerous studies that demonstrate a strong correlation between high viral loads (similar to those that we observe) and the presence of viable SARS-CoV-2 in viral cultures. We now briefly mention and cite these in the revised manuscript.

3- There is now a growing body of evidence of the persistence of replication-competent virus throughout the body months after start of an infection (10.1038/s41586-022-05542-y), and very recently that this persistence is strongly associated with higher risk of Long Covid (10.1016/S2213-2600(23)00142-X). We now mention these papers in the manuscript.

4- As we mentioned above, out of all persistent infections with three or more time points that had at least one pair of sequences with zero consensus difference, nearly half of them had at least one consensus change since the onset of infection suggesting that the virus is evolving at the consensus level as time progresses since the onset of infection.

5- As we mentioned above, no consensus change over a period of 1-3 months is not equivalent to no measurable within-host evolution. We now provide two examples where we find substantial sub-consensus viral activity in individuals with zero consensus change if enough time has passed since the onset of infection (see Supplementary Figure 7).

As I suggest above, I still find the absence of detectable consensus changes in the vast majority of the persistent infections perplexing. It is not out of line with what reviewer 1 commented (which was focused on how within host rates were undercounted for based on consensus changes). Given the available case reports (of which there are now many) show rates of consensus changes >2-3 per month, what are we to think then of majority with no consensus changes? This is somewhat important as it affects the overall incidence estimate.

There is no contradiction between the absence of consensus change during a persistent infection and the between-host evolutionary rate of ~2 changes per month. At the population level, the process of virus evolution is different because typically only one virus particle establishes a new infection (the “transmission bottleneck”), so any mutation on the genome of the viral particle being transmitted will instantly go to fixation in the recipient. In roughly $\frac{1}{3}$ to $\frac{1}{2}$ of transmission events does the transmitted virus have a mutation compared to the consensus in the source individual (i.e. the consensus in the source and recipient differ by 1 mutation). This gives a substitution rate of 0.5 per transmission, and if there are 4 transmissions per month, this gives a substitution rate of 2 per month, which is in line with the observed between-host substitution rates. Hence there are no contradictions between what is observed within individuals and at the population level. We believe this is a tangential point as the paper is not about between-host evolution, and, therefore, we believe this distracts from the key messages of the

paper. However, if the Editor deems it necessary, we could certainly add a paragraph on this.

We would also like to note that in the previous round, Referee 1 explained why, in the presence of viral replication but in the absence of strong adaptive selection, we should not expect to see consensus changes during the first weeks or months of infection, an argument that they also reiterated above. We therefore believe that Referee 2's comment that the lack of consensus changes within individuals is perplexing, is out of line with Referee 1's comment. To explain a bit further, Referee 1 pointed out that if there is limited within-host adaptation, *de novo* mutations will require a long time to increase in frequency during infection, and therefore to reach over 50% frequency that is required for the consensus to change.

Lines 404-406 - Elevated or accelerated (or "rapid" as is used in some places) within-host rates in persistent infection are by comparison to SARS-CoV-2 evolutionary rates along a chain of transmission, in which bottlenecks slow down the process of mutation accumulation. Somehow this should be clarified, lest the reader think that a persistent and non-persistent infection somehow have different within-host evolutionary rates (which may or may not be the case).

What makes the within-host evolution of a persistent infection different from that of the non-persistents who clear the infection more rapidly is that there will be more opportunities for the virus to acquire consensus changes over a longer time scale. We believe we have addressed this point in response to the previous comment and now briefly explain this in the introduction.

Reviewer 1 pointed out something that I missed on the initial review - the one individual with the large number of NS changes in the first 30 days (20 I think). This is really perplexing as it is 10x the typical evolutionary rate and much higher (perhaps 4-5x) the rates of mutation accumulation in immunocompromised hosts (from case reports). The most obvious explanation would be contamination during sequencing - but this seems less likely given the trajectories shown. Another possibility would be molnupiravir treatment, which increases the mutation rate. But the authors report they have no access to these data. Finally, APOBEC activity would be another possibility. It might be possible to sort this out by looking at the mutations and contexts closely. The current explanation - there's just rapid evolution - feels a little "hand wavy" for something that is so far from what has been reported.

We thank the Referee for highlighting this point. Incidentally, Referee 1 also suspected, based on the mutational patterns (many G->A mutations and few C->U mutations) that

we see in this individual, they might be under molnupiravir treatment. We have now mentioned this briefly in the results.

Referee #3 (Remarks to the Author):

The authors partially addressed my concerns. They have at least clarified what they did enough that one can determine the study design. This is a cohort study. They have taken a cohort study, reduced the study to look at those that were SARS-CoV-2 infected over the time period, assessed their main exposure (persistently infected or not persistently infected), and then followed them forward for their outcome (long COVID).

1) The appropriate checklist given this is a cohort checklist which can be found <https://www.strobe-statement.org/checklists/>

We have completed this checklist to the best of our ability as requested, reflecting the specific design of the survey and its primary objective which was to estimate daily swab positivity, as stated in the protocol (referenced in the manuscript). We would also note that the majority of the analysis included in the manuscript is genetic analysis of infections where multiple sequences were obtained, and that the Long Covid comparison which the reviewer focuses on is a small part at the end of the main text.

2) It is still not possible to assess potential bias due to incomplete surveys or samples. A flow chart which starts with all the infections that were identified between 2nd November 2020 to 15th August 2022 should be included (supplement is fine).

We have included a flow chart as suggested. However, given the underlying study design, which had the primary objective of estimating daily swab positivity rates under the constraints of total samples tested, it is simply not possible to provide some of the information requested. We respond to each point below.

The flow chart should specify who was not included in the analysis due to a lack of a follow-up sample or missing survey.

The design of the survey targeting a certain number of swab tests per month, which varied over calendar time, in order to meet its overall objectives makes this very challenging. Further participants were offered the option to only do one or five, rather than regular assessments. What we can assess is the distribution of time between assessments in those testing positive and negative - this was included in the previous Supplementary Table 4 and referred to in the main text. This shows that the distribution is similar between those testing positive and negative, and with failed tests. The flow diagram now explicitly states the numbers testing positive/negative and failed (only

percentages given in the previous revision). We have also explicitly added the number of participants with infections with Ct \leq 30 who were assessed for Long Covid.

The methods should also specify how that was handled as well. For example, what did the authors do if a participant was positive but failed to have a follow-up swab within the time necessary to determine that they were shedding persistently? For example, what if they were positive but didn't have another swab or their next swab was 45 days or more out and negative?

As requested we have clarified in the Methods that our identification of persistent infections was based on sequences obtained. Practically it is very difficult to see what else we could have done in a survey this size, given the constraints that sequencing was only attempted on samples with Ct \leq 30 and the fact that samples were taken independently of symptoms (a study strength) meaning that a reasonable proportion of infections were identified late in their time course (but unbiasedly).

It is certainly true that we may have missed some persistent infections because of this, and we have now noted this in the first paragraph of Discussion and Methods. This would lead to a dilution bias however, meaning that the impact on Long Covid was larger, not smaller.

What do the authors do if a participant was missing the follow-up survey to determine if they still had symptoms at 28 days or further out? Was missingness consistent between the two groups?

Current symptoms were completed at all assessments and were a mandatory field. Therefore if a swab test was taken, symptoms were completed. We have clarified this in the Methods but not added to the flow diagram as this is already complicated.

The authors provided information about the overall cohort. Still, here we really need information on the subcohort that was infected and should have been included in this analysis if the world were perfect (everyone completed every sample/survey etc), including how missing data was handled and who wasn't included due to missing data.

We have clarified our approach to missing data, as above, and tried to add as much information as possible to the flow diagram (now included as Supplementary Figure 1), but, given the design of the study in order to achieve its overall goal, and the fact that sequencing can only be attempted on samples with high viral load, it is simply not possible to say who should have been included if the world was perfect.

Referee #4 (Remarks to the Author):

This study uses genetic sequences from a large surveillance study of UK households to study persistent infections and reinfections with SARS-CoV-2. The combination of the data source and the approach is unique and generates a number of useful insights into the role of persistent infections in the evolution and epidemic dynamics of the virus.

My previous comments have been addressed adequately, and the methods and results are now described clearly. I have only a few typo corrections and a two minor suggestions about the description of the statistical methods.

1. (line 362) The "lower rate of reporting Long COVID" is actually a lower probability.

Amended.

2. (lines~357-366) The type of regression model used to adjust for possible confounders is never named. It would be good to specify that it was a logistic regression model either here or in lines~693-706.

The reviewer is correct that this is a binary logistic regression model and we have included this in the text as suggested.

3. (line 472) "supplementary table~4" needs to be capitalized, and "ref 45" should not be a superscript.

The text here is referring to supplementary table 4 of reference number 45, not of our paper. That is why it is not capitalised. We amended the issue with the reference. Thank you.

4. (lines~693-706) The description of the logistic regression model is clear, but no evaluation of goodness of fit (e.g., Hosmer-Lemeshow test or calibration belt) is described. It would be good to add this.

The model was solely used to control for measured confounders of the relationship between persistent positivity and Long Covid, which we selected on substantive rather than empirical grounds (i.e. using a causal inference approach). The model was not intended to be used to infer risk factors for Long Covid in a more general sense; goodness-of-fit measures such as the Hosmer-Lemeshow test are really only useful when the goal of modelling is prediction, which was not the case here. We have clarified this in the Methods.

Reviewer Reports on the Second Revision:

Referees' comments:

Referee #1 (Remarks to the Author):

The authors have addressed my remaining concerns and questions.

To address the concern by Ref 2 that the data presented does not allow to differentiate between persistent infection or persistent viral RNA production, the authors present the following arguments

(1) frequently observed rebounds in viral load would be difficult to explain without some form of replication. I imagine the underlying physiological reasoning is that the pool of cells producing viral RNA is expected to decrease in absence of de novo infection of cells. This makes sense to me.

One caveat to that argument is that some trajectories might look like rebounds due to noise. In order to classify an infection as persistent, the last time point needs to have a Ct < 30 to sequence and rule out reinfection. To detect a rebound, you need at least three time points, but the second time point could be any Ct and the conditioning on the last time point to be Ct < 30 will enrich for the samples that look like rebounds. I don't know to what extent variation in swabs, local variation in VL in the pharynx, variation in time of sampling all might generate variation in Ct values. Winnett et al, PNAS 2023 for example report dramatic differences from one day to the next and between saliva, anterior nasal swabs, and pharyngeal swabs. The present study samples only once a month, so variation of this sort could look like rebounds.

I am unsure to what extent remaining innate antiviral immune activation following the initial peak might suppress RNA production more around day 30 than around day 60. An effect like that might make the observation consistent with RNA production without replication. But my knowledge of immunology or molecular virology is insufficient to assess the plausibility of this.

(2) Numerous studies link high RNA viral load to the presence of infectious virus. It is a bit challenging to get a good sense of the VL at the later time points from Fig 3a&b. But they do tend to be around 25 and thus 100 to 1000-fold lower than the peak viral load (fig 3d). I don't think we can extrapolate from this to the likely presence of replicating virus.

(3) growing evidence for widely dispersed and persisting virus across the body. These papers certainly make the case that long term persistence is possible.

(4) accumulation of consensus differences in later time points. This is consistent with a shifting dynamics and ongoing evolution where variants carrying mutations eventually come to dominate. But shifting populations of RNA producing cells (and sampling differences) could also lead to differences in the consensus.

(5) examples of patients with the accumulation of minor variation. The examples shown in Supp Fig

7a&b show indeed considerable within host variation with additional iSNVs at later time points. It is a little surprising that these samples are already quite diverse at $t=0$ and this initial diversity seems to stay at more or less constant frequency.

My take is that the observations are more readily explained by (at least partially) ongoing replication of the virus, but I don't think this can be conclusively shown from the data available in this study.

Referee #2 (Remarks to the Author):

My comments have been adequately addressed.

Referee #3 (Remarks to the Author):

The authors have addressed some of my concerns, but some of their rebuttal raises concerns about the methods not being adequately described or indeed perhaps misrepresented in the manuscript.

For example, the manuscript states "All individuals aged two years and older from each household who provide written informed consent provide swab samples (taken by the participant or parent/carer for those under 12 years), regardless of symptoms, and complete a questionnaire at assessments, which occur weekly for the first month in the survey and then monthly."

However, in the rebuttal the authors state "The design of the survey targeting a certain number of swab tests per month, which varied over calendar time, in order to meet its overall objectives makes this very challenging. Further participants were offered the option to only do one or five, rather than regular assessments." These two descriptions seem very inconsistent.

The argument that the authors cannot provide data on whom they expected samples from because of the design seems questionable. And if this is indeed the case, then the design used is not appropriate for answering all of the questions that the authors have addressed here. It should be possible to provide a flowchart of participants and samples, after all the authors did arrive at a final dataset for the analysis.

In particular, I am concerned with the one about long COVID prevalence. Epidemiological studies, particularly large ones never have 100% complete data, however, providing information on data completeness is crucial for assessing bias.

Author Rebuttals to Second Revision:

Response to referees

We thank the editor and all the referees for their careful read of our manuscript and their helpful comments. We are very pleased that most of the referees' comments have now been addressed. Here we outlined our response to the remaining comments and how we revised the manuscript to reflect the new changes.

Editor comments

As we explained in our last decision letter, there were two outstanding issues that the referees were concerned about. The first was whether or not you could conclude that the persistent detection of viral RNA reflects persistent infection. Reviewer 2 is persuaded by your arguments, but we also asked reviewer 1 to weigh in on this point and as you can see below, reviewer 1 feels that although you are probably correct you can't conclude with certainty that there is ongoing replication. As such we would ask you to reframe the findings to focus on the detection of persistent viral RNA (not replicating virus or persistent "infection"). You can state that you believe the findings are more readily explained by ongoing replication but you should also state the caveats and limitations highlighted by reviewer 1. We feel a more appropriate title would be something like "Prevalence of persistent SARS-CoV-2 RNA in a community surveillance study" (note that titles have to be 75 characters or less, including spaces).

We agree that even though our results are most parsimoniously explained by the persistence of replication-competent SARS-CoV-2 in these individuals, we cannot definitively prove this for every case. However, we still believe that ongoing replication is by far the most likely mechanism for persistence, and is supported by subsequent analyses we have undertaken for a follow-on paper. Nevertheless, we are aware that there is an ongoing wider debate in the broader scientific literature, and we are, therefore, sympathetic to the Editor's request to reframe our findings in light of this important caveat.

As a title, may we suggest "Prevalence of persistent SARS-CoV-2 in a large community surveillance study"? We feel "prevalence of persistent SARS-CoV-2 RNA" in the title may not accurately reflect our findings, most importantly as this would imply we investigated the persistence of viral RNA in any and all positive PCR samples that persist for long durations, including those with Ct >30 and which were not sequenced; because they were not sequenced they were not included in our analysis. In addition, based on our ongoing research on characterising the within-host evolutionary dynamics of the virus in these individuals, we see further evidence that the evolutionary rate of the virus is aligned with that of continuously evolving (and most likely actively replicating)

virus populations. We feel this is a good compromise as we have removed the word “infection”, which implies replicating virus is unambiguously present, but also omits “RNA”, which may imply to many readers that the findings are due to the presence of non-replicating virus. This also enables us to reinsert the word “large” into the title to describe the community cohort; it really is very big (over 11 million swabs were taken) and so should be emphasised.

As well as the suggested title change, we have now explicitly clarified early on in the abstract and throughout the main text that any references to persistent infections indicates persistently replicating virus or RNA present at high titres over long periods. We hope these new changes are satisfactory. However, we are open to considering further modifications if the editor deems it necessary as it is not our intention to be misleading or to over-egg the significance of our findings. We also added the limitations of our findings highlighted by reviewer 1 in main text and discussion. We note that as a result of making this clarification in the abstract, our abstract currently has 230 words, 30 words beyond the standard word limit.

The second outstanding issue remains the association of persistent viral RNA with long-COVID symptoms. Reviewer 3 remains concerned about missing data and feels that if you cannot provide information on data completeness the study design may not be appropriate to answer the question. The reviewer also finds some of your responses confusing. This issue remains an obstacle to publishing the conclusions about long-COVID symptoms in Nature, and we would like to consider your response to these concerns (hopefully with some increased clarity about how missing data was handled).

We are grateful to the attention given by reviewer 3 on our Long-Covid analysis. We have now addressed these remaining concerns and believe that their comments helped us improve the clarity of our methods section.

Referee #3 (Remarks to the Author):

The authors have addressed some of my concerns, but some of their rebuttal raises concerns about the methods not being adequately described or indeed perhaps misrepresented in the manuscript.

For example, the manuscript states “All individuals aged two years and older from each household who provide written informed consent provide swab samples (taken by the participant or parent/carer for those under 12 years), regardless of symptoms, and complete a questionnaire at assessments, which occur weekly for the first month in the survey and then monthly.”

However, in the rebuttal the authors state “The design of the survey targeting a certain number of swab tests per month, which varied over calendar time, in order to meet its overall objectives makes this very challenging. Further participants were offered the option to only do one or five, rather than regular assessments.” These two descriptions seem very inconsistent.

We thank the referee for pointing this out.

Before addressing this specific comment, and others from the referee below, we would note that all versions of the study protocol are freely available on <https://www.ndm.ox.ac.uk/covid-19/covid-19-infection-survey/protocol-and-information-sheets>. These are now more explicitly referenced and clarified in the methods section of the manuscript.

Further the sheer size and scale of the study, and the fact that the study was set up in April 2020 when the full implications of the pandemic were simply unknown, and then ran for almost 3 years under substantial but varying funding envelopes, meant that we did need to make some trade-offs. During the 3 years that the survey ran before it was paused, over 11 million swabs were taken of which around 205,000 were RT-PCR positive (1.8%) and over 125,000 of these produced high-quality sequences.

Whilst completely appreciating the referee’s concerns, and the importance of clarity (addressed below), the key substantial strength of this study, in comparison with virtually every other study of infections in a true community-based sample, is the fact that sampling was done independently of symptoms or any other patient-level characteristics. The fact that there was some variability in timing between these assessments, pre-specified in the ethically approved protocol as is standard in all observational studies and clinical trials, in no small part due to the complexities of managing a field force of 2000-3000 study workers across the entirety of the UK, is inevitable - but it is an extrinsic source of variation. As the manuscript is already quite long, we have not added anything further on this background directly to the text (although have added/amended text to address the referee’s specific points, as described below). However, we would be happy to do so if the referee or editor felt it was helpful.

In terms of the specific comment above, the first statement was in reference to the sampling protocol for each consenting participant *who agreed to ongoing repeat sampling at enrollment* (approximately 98% of consenting participants agreed to this, see flow diagram). Once recruited, these individuals were swabbed approximately

weekly for the first month and then monthly thereafter. The statement “Further participants were offered the option to only do one or five, rather than regular assessments...” refers to the approximately 1% of participants who opted to give only one sample, and the approximately 1% who opted to give samples for the first month only. As well as in the methods, this was included in the flow chart. The reason for this design (clearly laid out in all the protocols from version 1.0, see <https://www.ndm.ox.ac.uk/covid-19/covid-19-infection-survey/protocol-and-information-sheets>) is because at the start of the pandemic in April 2020, there was substantial concern that many people would not want to commit to long-term sampling, but that data from a single or even five assessments would still be very useful.

The second statement that “The design of the survey targeting a certain number of swab tests per month...” is in reference to the overarching target for the number of swabs taken per month, not the sampling protocol, since the overall objective of this study was surveillance, i.e. to provide a certain degree of precision around estimates of positivity at national and regional levels. This overall objective determined the funding envelope, and hence this target did both increase and decrease on occasions as the survey progressed (see <https://www.ndm.ox.ac.uk/covid-19/covid-19-infection-survey/protocol-and-information-sheets>) in order to meet the government’s current needs regarding precision of national and regional estimates. To meet the target number of swabs, additional households were recruited; also additional households were recruited to replace households leaving the survey, predominantly through moving house (since eligibility was determined by the original address which was sampled, not the participants themselves). Once recruited, participants would follow the sampling protocol above, using only the protocol prespecified windows around planned assessments to match survey targets (coupled with varying the rate of inviting new households to join the study). In other words, the aim was to sample all currently recruited participants *according to the same sampling protocol* as pre-specified in the protocol in order to achieve the target number of swabs for the survey, rather than use the target number of swabs to determine the assessment schedule. We have now clarified this in the manuscript. In addition, we have clarified throughout that assessments were approximately monthly, rather than monthly. In the section “ONS COVID-19 Infection Survey”, we now write:

“The survey offered participants the option of only having one enrollment assessment (taken by ~1%), or weekly assessments for only one month (taken by ~1%; see **Extended Data Figure 1**). All other enrolled participants (~98%) were assessed weekly for the first month of their enrollment in the survey and then approximately monthly (originally for one year; all such participants were approached for re-consent for ongoing follow-up beyond one year). The survey had rolling recruitment in order to meet its

target for taking a certain number of swabs from the population each month, but in practice most recruitments occurred between September and December 2020 (**Source Data 2**; also see supplementary table 4 in ref ²). The rolling recruitment enabled the study to achieve its overall sample numbers (required to address its surveillance objectives) whilst accounting for participants withdrawing from the study, for example when they moved house (since eligibility was based on the originally sampled address). As is standard, the protocol also allowed a 14-day window around the approximately monthly assessments (shifting any following assessments to avoid swabbing participants again at very short (and variable) notice); crucially, assessments were not missed in order to meet survey targets.”

The argument that the authors cannot provide data on whom they expected samples from because of the design seems questionable. And if this is indeed the case, than the design used is not appropriate for answering all of the questions that the authors have addressed here. It should be possible to provide a flowchart of participants and samples, after all the authors did arrive at a final dataset for the analysis. In particular, I am concerned with the one about long COVID prevalence. Epidemiological studies, particularly large ones never have 100% complete data, however, providing information on data completeness is crucial for assessing bias.

As explained above, as standard and as stated in all the protocols on <https://www.ndm.ox.ac.uk/covid-19/covid-19-infection-survey/protocol-and-information-sheets>, the protocol allowed windows around scheduled assessments - allowing flexibility both for study workers (2000-3000 visiting participants all over the UK) and for participants being visited at home. If an assessment was shifted, the following assessment was also shifted to avoid swabbing participants again at very short (and variable) notice. We disagreed with the referee’s assertion that the design is not suitable to address the objectives of the current analysis because any variation in the study sampling schedule was extrinsic to the participant, that is, was independent of patient characteristics and symptoms in particular. In contrast, other designs based on testing programmes, on which almost all literature is based given the complexity and cost of conducting a study such as ours, have enormous potential for bias from decisions around whether and when to test.

As shown in the flow diagram provided with the previous revision (extended figure 1), apart from the ~2% of participants providing consent for only one assessment or five assessments spanning one month, the rest of the participants were swabbed for a much longer period, covering both before and after positive PCR tests. The previously provided flow diagram also included information about the median IQR time between assessments where a positive or negative swab result was obtained (28 (27-42) days),

showing that the majority of assessments were separated by <6 weeks. We had also provided the detailed and full distribution of time between assessments by swab result overall and in those testing positive in the previous supplementary table 4 (now renamed Source Data 3; please note we identified that some swabs which had been called Void by the lab but where they had provided (predominantly high) Ct values had inadvertently been included in these summaries – we will remove these from the table once we clear the data from the ONS with minimal impact on the distributions). This previously provided flow diagram also showed the number and percentage of those without persistent infection (but with other characteristics, specifically Ct and testing dates, matching those with persistent infection) who were assessed for Long Covid at least 12w (93%) and 26w (86%) after their infection.

In order to provide additional information, we will calculate the number and percentage of assessments during the study period and after its introduction at which the Long Covid question was completed, and we will add this to the Methods. The reason we have not added these numbers immediately is because we need clearance to export and publish this data from the ONS and this can take a few weeks, but we will forward them as soon as we have them. We are not sure what other information on the assessment of Long Covid would be useful in addition to what is now presented, but would be happy to try to calculate any specific quantities that could be useful.

One possibility, noting that this was not how the protocol was implemented, is that we could provide the number of assessments that would have been counted as missed had the protocol been implemented following a fixed schedule from the original household enrolment date. We have not currently added this text as it is hypothetical and would need a lot of explanation and may be more confusing than it is illuminating. However, we could add it if desired.

Reviewer Reports on the Third Revision:

Referees' comments:

Referee #3 (Remarks to the Author):

Pending the addition of the information that the authors offered to provide in the response to reviewers, my concerns have been addressed.